

# ISMIP6-based Antarctic Projections to 2100: simulations with the BISICLES ice sheet model

James F. O'Neill[1], Tamsin L. Edwards[2], Daniel F. Martin[3], Courtney Shafer[4], Stephen L. Cornford[5], Hélène L. Seroussi[6], Sophie Nowicki[4], and Mira Adhikari[2]

[1]University of Exeter
[2]King's College London
[3]Lawrence Berkeley National Laboratory
[4]University at Buffalo, State University of New York
[5]University of Bristol
[6]Dartmouth College

**Correspondence:** James O'Neill (j.oneill2@exeter.ac.uk)

**Abstract.** The contribution of the Antarctic ice sheet is one of the most uncertain components of sea level rise to 2100. Ice sheet models are the primary tool for projecting future sea level contribution from continental ice sheets. The Ice Sheet Model Intercomparison for the Coupled Model Intercomparison Phase 6 (ISMIP6) provided projections of the ice sheets contribution to sea level over the 21st century. It quantified uncertainty due to ice sheet model, climate scenario, forcing climate model and

uncertain model parameters. We present simulations following the ISMIP6 framework with the BISICLES ice sheet model, alongside new experiments extending the ISMIP6 protocol to more comprehensively explore uncertain ice sheet processes. These results contributed to Antarctic projections of Edwards et al. (2021), which formed the basis of sea level projections for the Sixth Assessment Report of the Intergovernmental Panel on Climate Change (AR6). The BISICLES experiments presented here show the important interplay between surface mass balance forcing and ocean driven melt, with high warming, high

accumulation forcing conditions leading to mass gain (negative sea level contribution) under low sensitivity to ocean driven melt. Conversely, we show that when sensitivity to ocean warming is high, ocean melting drives increased mass loss despite high accumulation. Finally, we show that collapse of ice shelves due to surface warming increases sea level contribution by 25 mm for both moderate and high sensitivity of ice shelf melting to ocean forcing tested.

## 1 Introduction

The Antarctic and Greenland ice sheets are the third largest contributor to global mean sea level (GMSL) behind thermosteric changes and mountain glaciers (Palerme et al., 2017; Horwath et al., 2022), dominating the sea level change of 20.0 cm between 1901 and 2018 (Fox-Kemper et al., 2021). In recent decades, ice sheet mass loss has made up a growing proportion of sea level rise, which averaged $3.64 \pm 0.26$ mm yr$^{-1}$ between 2003 and 2016 (Horwath et al., 2022). From 1992 to 2020, the Antarctic ice sheet (AIS) contributed $7.4 \pm 1.5$ mm to global mean sea level rise (Otosaka et al., 2023). Although Antarctica was a smaller

source of GMSL between 1993 and 2016 than other land ice sources and land water storage (Horwath et al., 2022), evidence of past volume and dynamics suggest that the ice sheet could become a significant source of GMSL in a warming climate





(DeConto et al., 2021; Lowry et al., 2021; Edwards et al., 2019). To date, mass loss in Antarctica has been dominated by ice streams and marine terminating glaciers responding to warm ocean currents eroding ice shelves in the Amundsen Sea sector (Shepherd et al., 2018; Rignot et al., 2019), with changes in ocean currents linked to anthropogenic warming-driven changes to

wind regimes (Holland et al., 2019). Along with some East Antarctic basins, the West Antarctic ice sheet is vulnerable to ocean driven instabilities as grounding lines retreat into over-deepened subglacial basins (Schoof, 2012; Weertman, 1974; Thomas, 1979). Around 23 m of sea level equivalent ice rests on bedrock below sea level (Morlighem et al., 2020).

Under anthropogenic warming, destabilisation of marine basins could drive accelerating Antarctic GMSL contribution to 2100 and beyond (Schlegel et al., 2018; Bulthuis et al., 2019; Lowry et al., 2021; Edwards et al., 2019; DeConto and Pollard,

2016; Golledge et al., 2015; Ritz et al., 2015). Moreover, dynamic instabilities amplify uncertainty in future sea level projections (Robel et al., 2019). Alongside this ocean-driven retreat under anthropogenic warming, increased Antarctic surface mass balance has the potential to mitigate the ice sheet's sea level contribution. Warmer air temperatures over Antarctica can increase precipitation, driving increased surface mass balance under the cold, low melt conditions of the ice sheet (Frieler et al., 2015; Palerme et al., 2017). Over the course of the 20th century, increased precipitation offset  10 mm of AIS-sourced GMSL

(Medley and Thomas, 2019).

To better capture Antarctic contribution to sea level, ice sheet models have been developed to represent a greater range of interactions and dynamic processes, at higher resolution than ever before, over the past few decades (Pattyn et al., 2017). However, differences in process representation, model physics, spatial discretisation and initialisation (Seroussi et al., 2019) mean that different ice sheet models project different AIS responses to the same climate boundary conditions (Edwards et al.,

2014; Bindschadler et al., 2013). The Ice Sheet Model Intercomparison Project for the Coupled Modeled Intercomparison Projects Phase 6 (CMIP6), ISMIP6 (Nowicki et al., 2016), builds on previous multi-model ensemble efforts (e.g. Edwards et al. 2014; Bindschadler et al. 2013) to better characterise uncertainty in projected future GMSL from the Greenland and Antarctic ice sheets arising from both ice sheet and global climate models. (Nowicki et al., 2016).

ISMIP6 explores the role of ice sheet models that differ in approximations of ice physics, basal sliding, model resolution

and initialisation approaches, providing consistent climate forcings and suggesting an ice shelf basal melt parameterisation for projection experiments (Nowicki et al., 2016). By including a range of ice sheet models with similar boundary conditions, ISMIP6 quantifies the range of modelled sea level projections due to ice sheet model choice.

Results of ISMIP6 Antarctic ice sheet experiments forced with Coupled Model Intercomparison Project Phase 5 (CMIP5) climate models are described by Seroussi et al. (2020), who find a range of -7.8 cm to 30.0 cm sea level equivalent (SLE)

contribution from Antarctica from 2015 to 2100 under a very high emissions scenario (RCP8.5) compared with experiments under constant climate conditions. Under a low emissions scenario (RCP2.6) with two CMIP5 models, an average additional mass loss of 0.0 to 3.0 cm is found compared with simulations under modern climate (Seroussi et al., 2020). Comparing these results with a further set using next generation CMIP6 climate model forcings, Payne et al. (2021) find a limited difference between projections grouped by generation of CMIP climate model. This is attributed to the complexity of interactions between

the AIS and the climate system, with warming-linked surface mass balance increases offsetting ocean melt driven mass loss in some cases (Payne et al., 2021). Whilst CMIP6 models generally simulate more warming than CMIP5 models, both ocean





melting and surface mass balance are enhanced in CMIP6, so that sea level contribution does not differ significantly by CMIP generation (Payne et al., 2021). BISICLES ISMIP6 Antarctic experiments were used in the ISMIP6 synthesis and sensitivity tests of Edwards et al. (2021). However, with the exception of experiments for the model initialisation intercomparison exercise

(InitMIP) (Seroussi et al., 2019), ISMIP6 BISICLES simulations have not yet been presented in detail ((Edwards et al., 2021)).

We present a set of 19 simulations (18 projections and a control) from the BISICLES ice sheet model following the ISMIP6 protocol for future projections of the Antarctic ice sheet. Our simulations follow the design for Tier 1, 2 and 3 experiments (Nowicki et al., 2016). Tier 1 were core ISMIP6 experiments, using climate forcing derived from the highest skill models in Barthel et al. (2020), exploring scenario dependence and sensitivity to shelf collapse and ice shelf basal melt sensitivity

(Nowicki et al., 2016). These experiments were mandatory for inclusion in ISMIP6. Tier 2 experiments explore a wider range of models assessed in Barthel et al. (2020) from the CMIP5 ensemble, as well as CMIP6 models based on availability. Tier 3 experiments provide a more in-depth exploration of the role of ocean sensitivity in modelled AIS evolution to complement Tier 1 (Nowicki et al., 2016).

We also explore the relationship between ocean melt and ice shelf collapse through additional sensitivity experiments. The

Tier 1-3 experiments contribute to the ISMIP6 effort by adding another Antarctic ice sheet model to the ensemble, while the additional sensitivity experiments target uncertainties in the synthesis by Edwards et al. (2021): by testing for interactions between uncertain parameters.

Here we present the BISICLES model set-up and experimental design (Section 2) and results of the 19 ice sheet model experiments (Section 3). We then discuss the role of different modelling choices on Antarctic contribution to sea level, compare

BISICLES to other ISMIP6 ice sheet models, and finally discuss limitations of our approach (Section 4).

## 2    Methods

### 2.1    BISICLES

BISICLES is a block-structured, finite volume, L1L2 physics ice sheet model with adaptive mesh refinement (Cornford et al., 2013, 2015, 2016). For these simulations, we use the BISICLES_B model set up as in Seroussi et al. (2019). All simulations are

run with a base resolution of 8 km, with 3 levels of refinement to reach a finest mesh resolution of 1 km. The model domain at the coarsest level covers a grid of 768 x 768 cells. We use the subgrid friction interpolation scheme described in Cornford et al. (2016). This allows for finest resolution of 1 km at the grounding line and in regions of fast flowing ice, adequately capturing grounding line dynamics compared with higher resolution simulations where the subgrid friction scheme is not used(Cornford et al., 2016).

Basal traction is calculated using a pressure-limited Weertman-Coulomb type law (Tsai et al., 2015) with m=1/3 and a Coulomb friction coefficient of 0.5. Basal traction coefficients and the ice damage coefficient are estimated using an inversion approach to minimise the mismatch between observed and modelled ice speed (Cornford et al., 2015), and are held constant throughout the simulations. Ice temperature is from Pattyn et al. (2010), who simulated ice sheet temperature with a 3D thermo-mechanical ice sheet model, and is fixed throughout the simulations. Whilst BISICLES uses a depth integrated momentum





balance equation, effective viscosity accounts for 3D temperature of the ice sheet, so a 3D temperature profile is used. In the
experiments presented here, the calving front is fixed. All simulations are initialised from a short relaxation run, as in previous
BISICLES studies (Cornford et al., 2016). Whilst the ISMIP6 analysis period is from 2015 to 2100, our simulations start in
2010 and use the ISMIP6 forcing anomalies provided, which cover the period 1995-2100.

Ice sheet contribution to sea level is equal to change in volume above floatation (VAF) in the absence of bedrock deformation
- a process we do not include. Volume above floatation is the volume of ice sheet that is not below sea level or hydrostatic
equilibrium, and is therefore not already displacing ocean water. To calculate sea level contribution, i.e. change in VAF in
metres sea level equivalent (m SLE) for the modern ocean, we distribute sea level equivalent change in VAF over an ocean area
of $3.625 \times 10^{14}$ m$^2$ (Gregory et al., 2019), with ocean density 1028 kg m$^{-3}$ and ice density 918 kg m$^{-3}$.

### 2.2   Ocean and atmosphere forcing

In order to facilitate a consistent approach across participating modelling groups, ISMIP6 provides surface mass balance and
ocean thermal forcing data from a subset of CMIP5 and CMIP6 climate models. The selection process for core ISMIP6
experiment model forcings is outlined in Barthel et al. (2020). Core experiments use CMIP5 model outputs to provide climate
forcings, which are chosen based on skill in simulating atmospheric and ocean variables compared with observations (Barthel
et al., 2020), whilst sampling a diverse subset of models in terms of simulated climate by the end of the 21st century. The
selection of CMIP6 boundary condition models was based on availability. We use one CMIP6 model in our simulations -
CNRM-CM6-1, as forcing data were available for both low (SSP1.26) and high (SSP5.85) emissions scenarios. We could
therefore explore scenario dependence across a wider range of GCMs, and across CMIP generations.

To promote a consistent approach to basal melt forcing across ice sheet models, most participating groups used a prescribed
basal melt parameterisation (Jourdain et al., 2020; Nowicki et al., 2020). This parameterisation describes the relationship
between basal melting, $m$, and ocean thermal forcing, $TF$. BISICLES implements the "non-local" basal melt rate parameteri-
sation. Basal melt anomalies, relative to the initial melt forcing, are applied for each simulation year. The non-local basal melt
parameterisation captures the melt-induced cavity scale circulation changes that drive shelf melt Jourdain et al. (2017), as well
as the local influence of stratification, and compares favourably to coupled ice sheet ocean models in idealised experiments
(Favier et al., 2019). A more comprehensive description can be found in (Jourdain et al., 2020). It is restated here:

$$
\begin{aligned}
m(x,y) =& \gamma_0 \times \left( \frac{\rho_{sw} C_{pw}}{\rho_i L_f} \right)^2 \\
& \times (TF(x,y,z_{draft}) + \delta T_{sector}) \\
& \times |\langle TF \rangle_{draft \in sector} + \delta T_{sector}|,
\end{aligned} \tag{1}
$$


where $\rho_i$ and $\rho_{sw}$ are the densities of ice (918 kg m$^{-3}$) and sea water (1028 kg m$^{-3}$) respectively; $L_f$ is the fusion latent heat
of ice ($3.3 \times 10^5$ J kg$^{-1}$); and $C_{pw}$ is the specific heat of sea water (3974 J kg$^{-1}$ K$^{-1}$). Thermal forcing $TF$ is calculated at the





ice-ocean interface, while $\langle TF \rangle$ is averaged over each of the 16 Antarctic sectors. Figure 1 shows thermal forcing averaged over the surface ocean (500 m) from 2015 to 2100 for the GCMs used here.

The basal melt parameter, $\gamma_0$, is calibrated using two sets of melt estimates to span a wide range of possible sensitivities of the ice shelves to basal melt. The two sets of melt estimates are based on total Antarctic basal melt (Depoorter et al., 2013; Rignot et al., 2013) (*MeanAnt*) and melting at the grounding line of Pine Island Glacier (*PIGL*), respectively (Jourdain et al., 2020). In all, six values of $\gamma_0$ are provided (Table 1), corresponding to the $5^{th}$, $50^{th}$ and $95^{th}$ percentiles of the distribution for the low (*MeanAnt*) and high (*PIGL*) melt tuning. Five $\gamma_0$ values are sampled in the simulations presented here (Table 2) - we
did not use $PIGL_{5th}$.

| Calibration | $5^{th}$ | Median | $95^{th}$ |
|---|---|---|---|
| *MeanAnt* | 9,620 | 14,500 | 21,000 |
| *PIGL* | 88,000 | 159,000 | 471,000 |

**Table 1.** Calibrated values of basal melt parameter, $\gamma_0$, in m yr$^{-1}$

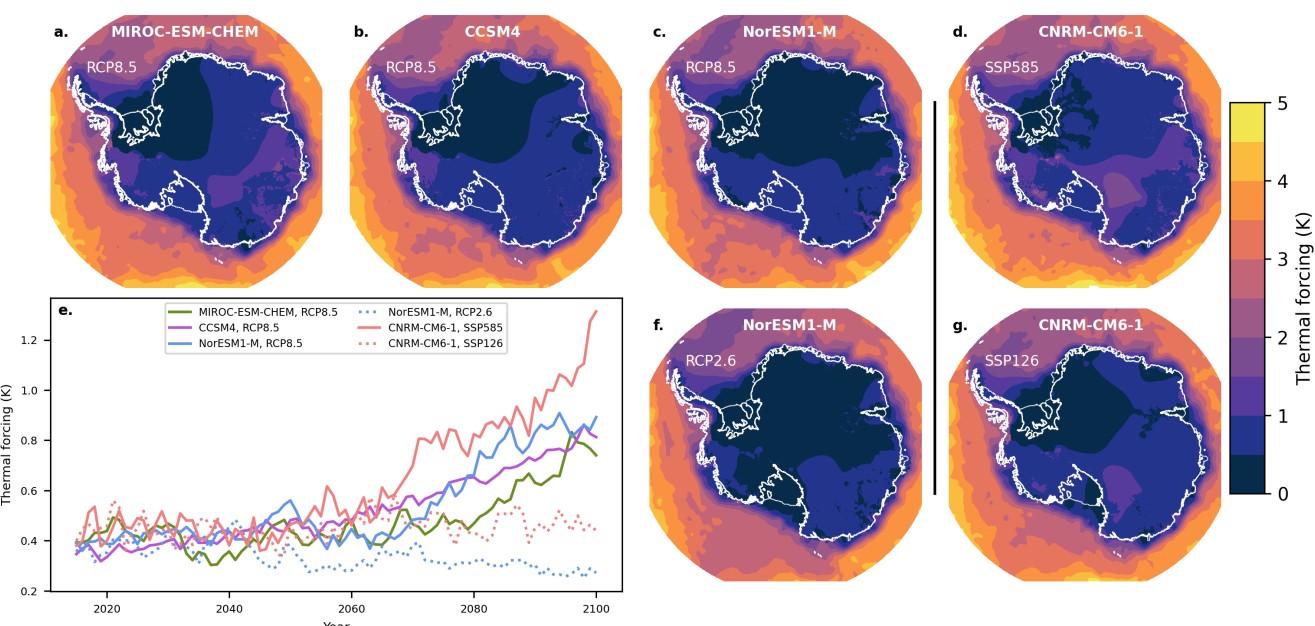

**Figure 1.** Thermal forcing averaged over the upper ocean (500 m) (**a**,**b**,**c**,**d**,**f**,**g**) from 2015 to 2100 for each of the climate models and emissions scenarios. Subplot **e** shows Antarctic mean annual upper (500 m) ocean thermal forcing from 2015 to 2100. Black vertical line separates CMIP5 models (under RCP scenarios) from the CMIP6 model (SSP scenarios).

For the ice/ atmosphere interface , surface mass balance anomalies were provided directly from GCMs relative to a January 1995 to December 2014 reference period (Nowicki et al., 2016). The anomalies were added to a baseline surface mass balance





from Arthern et al. (2006). This approach does not account for the evolving topography of Antarctica over the simulation period. Average surface mass balance anomalies for 2015 to 2100 are shown in Fig. 2.

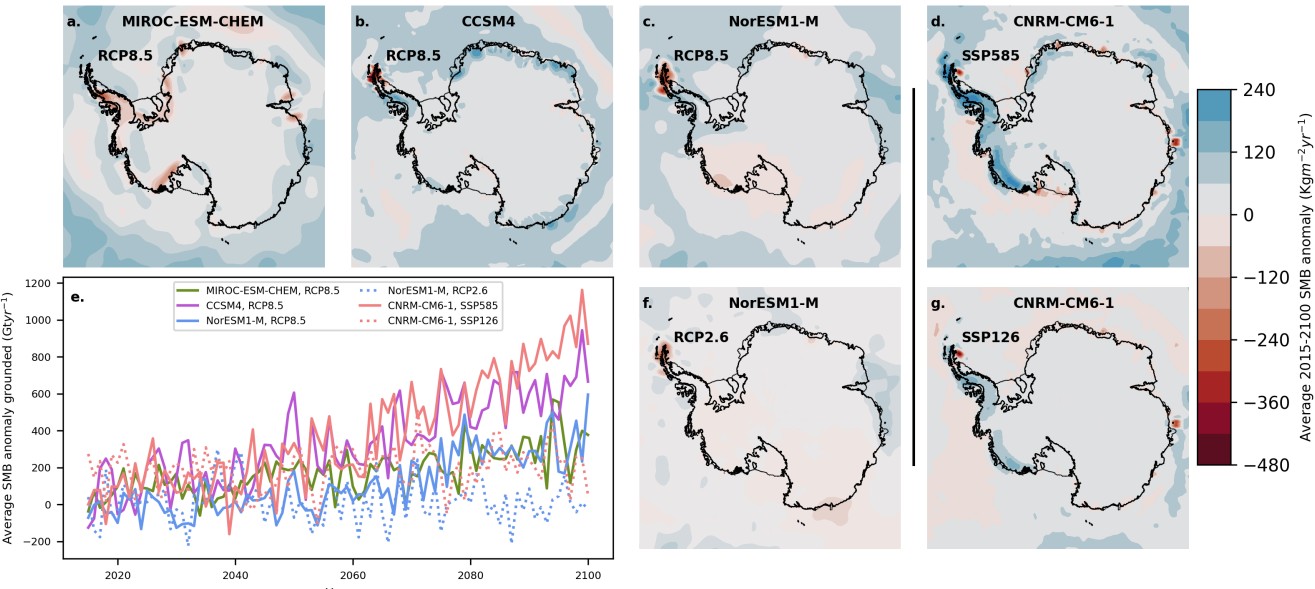

**Figure 2.** Surface mass balance anomaly (relative to 1995-2014) averaged from 2015 to 2100 (subplots **a,b,c,d,f,g**). Subplot **e** shows Antarctic mean annual surface mass balance anomaly from 2015 to 2100. Black vertical line separates CMIP5 models (under RCP scenarios) from the CMIP6 model (SSP scenarios).

Surface melt water can enhance propagation of crevasses in the ice shelf, driving weakening and eventual collapse(Scambos et al., 2009). However, inclusion of melt-driven hydrofracture and subsequent shelf collapse is a relatively recent innovation in ice sheet models (Pollard et al., 2015), and is not directly implemented in those participating in ISMIP6. ISMIP6 therefore provide time-dependent masks of ice shelf collapse to represent surface melt-enhanced shelf disintegration. The masks are derived from atmospheric forcing projections: if surface air temperature-driven melt of 725 $mma^{-1}$ water equivalent persists
for 10 years, the ice shelf is removed (Trusel et al., 2015; Seroussi et al., 2020).

We explore the impact of shelf collapse with two pairs of experiments (Table 2). Both sets of 'collapse on' simulations use the same shelf collapse mask, i.e. derived from the same climate model projections (CCSM4). In these experiments, shelf collapses progress southward during the experiment, from the Antarctic Peninsula towards the South pole.

## 2.3 Control simulation

The ISMIP6 protocol subtracts a control simulation from each projection simulation, to remove model drift (Nowicki et al., 2016). Our control simulation uses the baseline modern surface mass balance field from Arthern et al. (2006) to which the anomaly is added in projections. Basal melting is applied such that localised thickening - as a result of ice advection or surface





mass balance - is removed. Basal melt driven by ocean thermal forcing is not applied, and accumulation onto the lower surface is not permitted (see BISICLES_B in Seroussi et al. (2019)). Whilst shelves can thin locally due to advection of ice out of grid

cells, this holds the ice shelves close to to their initial geometry.

## 3 Results

### 3.1 Control simulation

From 2015 to 2100, the control simulation lost 50,149 Gt of total mass, of which 19,220 Gt was above sea level, contributing 60 mm to sea level (Fig. 3 (c)). The ice sheet area decreased by $6.9 \times 10^3$ km$^2$, while the floating area increased by $64.6 \times$

$10^3$ km$^2$.

Thinning occurs over large regions of the Amundsen Sea sector, with some grounding line retreat at Thwaites glacier (Fig. 3 (a)). Major ice shelves (Ross, Ronne-Filchner and Amery) also thin, along with their tributary ice streams. However, thinning of Lambert Glacier is less pronounced than in some ice streams on the Siple coast or those feeding the Ronne-Filchner shelf, consistent with a limited response of this catchment to shelf thinning in previous studies e.g. Gong et al. (2014). In East

Antarctica, ice streams at the margins of Victoria Land, Wilkes Land and Queen Maud land all undergo thinning in the control experiment, as do ice shelves in the Peninsula along with grounded ice abutting the George VI ice shelf.

The most pronounced ice stream speed up in the control simulation occurs in the Thwaites glacier and its ice shelf (Fig. 3 (b)), in response to grounding line retreat. By contrast, Pine Island glacier slows down between 2015 and 2100 in the control run. Along the Siple coast, Whillans ice stream (Ice Stream B) accelerates between 2015 and 2100, with grounding lines in this

sector undergoing modest retreat (Fig. 3 (a)). Overall, outer edges of major ice shelves slow down over the simulation period, with the exception of some ice shelves on the Dronning Maud Land and the West Ice shelf. In these latter sectors, localised grounding line retreat is associated with speed up of ice across the grounding line and out to the shelf edge.

### 3.2 Projected sea level contribution

Projected changes in sea level contribution during 2015-2100 across the 18 experiments, relative to the control simulation, vary

between -53 mm and 125 mm SLE (Table 2; Fig. 4). The majority show net mass loss. Five gain mass relative to the control, i.e. more mass is gained through accumulation than lost through basal melt and dynamic thinning: all of these use the lower basal melt ($MeanAnt$) parameterisation, and are forced by two of the four GCMs (CCSM4 from CMIP5 and CNRM-CM6-1 from CMIP6), and four of the five are under very high emissions scenarios (RCP8.5 or SSP5-8.5).

### 3.3 Projected changes in ice area

Grounded ice sheet area changes are shown in Fig. 5, grouped by GCM forcing. All simulations lose grounded area by 2100, with the exception of those forced by NorESM1-M under low emissions (RCP2.6) using the high basal melt parameterisation ($PIGL_{95}$), though under $PIGL_{50}$ and $PIGL_{95}$ the decrease is not monotonic. Perhaps counter-intuitively, initial grounded



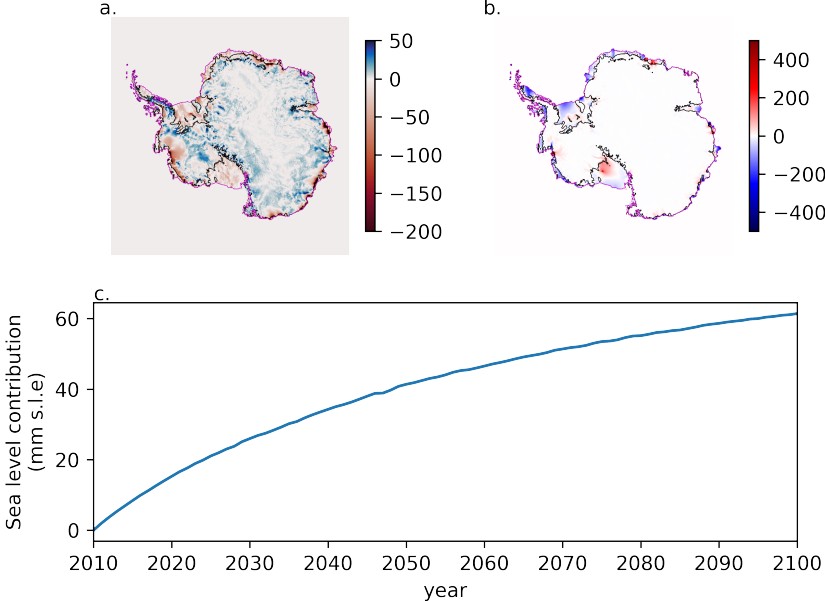

**Figure 3.** Control simulation (2010-2100). Subplot **a** shows the thickness change in metres, and subplot **b** shows the change in ice speed in m yr$^{-1}$. Subplot **c** shows the sea level contribution for the control simulation.

area increases with greater basal melt sensitivity to thermal forcing (i.e., higher values of $\gamma_0$: darker colours in Fig. 5). The differences in experiments between $MeanAnt$ percentiles are small because the $\gamma_0$ values are relatively similar (Table 1).
However, high basal melt sensitivity ($PIGL$) experiments decrease in grounded area much more quickly, generally to smaller final values than the $MeanAnt$ experiments despite their larger areas in 2015.

In contrast, floating ice area is larger at 2100 compared with 2015 for all experiments, with the exception of those with ice shelf collapse (CCSM4: triangles in Fig. 6) and the experiment forced with NorESM1-M under RCP2.6 using the high basal melt parameterisation ($PIGL_{95}$). This response - i.e. reduced grounded ice sheet area and increased floating area - is consistent
with grounding line retreat and loss of volume above floatation, with fixed front calving maintaining the shelf edge position so floating area increases.

### 3.4 Regional sea level contributions

To explore the distinct responses of the West Antarctic ice sheet (WAIS), East Antarctic ice sheet (EAIS) and the Antarctic Peninsula (AP) to perturbed boundary conditions and basal melt sensitivity, we partition sea level equivalent mass change for
these three regions. For WAIS, sea level contribution ranges from 103 mm SLE to -22 mm SLE . All but two experiments project a WAIS contribution to sea level rise at 2100. Projected sea level contribution for the EAIS ranges from 52 mm SLE rise to a sea level fall of 45 mm SLE. Volume above floatation increases to 2100 for the EAIS in more experiments compared



| Experiment | Scenario | GCM | $\gamma_0$ (m a$^{-1}$) | $\gamma_0$ percentile | Collapse | Sea level contribution (mm SLE) |
|---|---|---|---|---|---|---|
| exp05 | RCP8.5 | NorESM1-M | 14,477 | $MeanAnt_{50}$ | OFF | 31 |
| exp06 | RCP8.5 | MIROC-ESM | 14,477 | $MeanAnt_{50}$ | OFF | -2 |
| exp07 | RCP2.6 | NorESM1-M | 14,477 | $MeanAnt_{50}$ | OFF | 38 |
| exp08 | RCP8.5 | CCSM4 | 14,477 | $MeanAnt_{50}$ | OFF | -45 |
| exp09 | RCP8.5 | NorESM1-M | 21,005 | $MeanAnt_{95}$ | OFF | 39 |
| exp10 | RCP8.5 | NorESM1-M | 9,619 | $MeanAnt_5$ | OFF | 23 |
| exp12 | RCP8.5 | CCSM4 | 14,477 | $MeanAnt_{50}$ | ON | -20 |
| exp13 | RCP8.5 | NorESM1-M | 159,188 | $PIGL_{50}$ | OFF | 82 |
| expD52 | RCP8.5 | NorESM1-M | 471,264 | $PIGL_{95}$ | OFF | 91 |
| expD53 | RCP8.5 | MIROC-ESM | 159,188 | $PIGL_{50}$ | OFF | 71 |
| expD55 | RCP8.5 | MIROC-ESM | 471,264 | $PIGL_{95}$ | OFF | 121 |
| expD56 | RCP8.5 | CCSM4 | 159,188 | $PIGL_{50}$ | OFF | 31 |
| expD58 | RCP8.5 | CCSM4 | 471,264 | $PIGL_{95}$ | OFF | 102 |
| expT71† | RCP2.6 | NorESM1-M | 159,188 | $PIGL_{50}$ | OFF | 62 |
| expT73† | RCP2.6 | NorESM1-M | 471,264 | $PIGL_{95}$ | OFF | 57 |
| expTD58† | RCP8.5 | CCSM4 | 471,264 | $PIGL_{95}$ | ON | 125 |
| expB6 | SSP5-8.5 | CNRM-CM6-1 | 14,477 | $MeanAnt_{50}$ | OFF | -53 |
| expB7 | SSP1-2.6 | CNRM-CM6-1 | 14,477 | $MeanAnt_{50}$ | OFF | -17 |

**Table 2.** Experiment list with projected sea level contribution relative to the control simulation from 2015 to 2100. The †symbol indicates new experiments that were not part of the ISMIP6 protocol.

with WAIS – with 6 projections of sea level fall at 2100. Projections of Antarctic Peninsula sea level contribution range from 9 mm SLE to -6 mm SLE. For this region, 10 experiments project increased volume above floatation relative to control. We note that the CMIP6 CNRM-CM6-1 forced simulations result in sea level fall for both scenarios and all regions.

To further partition SLE ice sheet mass change, results are presented for 16 drainage basins. Following the ice sheet mass balance inter-comparison exercise (IMBIE) assessment (Shepherd et al., 2018), Antarctica is initially divided into 18 sectors - with each of the major ice shelves (Ross and Filchner-Ronne) bisected by a sector boundary. Then, to reflect the connectivity of water masses under the major ice shelves and avoid unphysical melt discontinuities imposed by sector boundaries within cavities, the two sectors feeding the Ross and the Ronne-Filchner ice shelves are merged to give 16 basins in total (Jourdain et al., 2020). The basins are used to derive basin scale parameters in the basal melt parameterisation, and to extrapolate ocean conditions under ice shelves in the ocean data preparation (Jourdain et al., 2020).

In terms of sectoral sea level contribution, the Totten sector (4) has the largest sea level contribution of any sector for the most runs (n=9) (Fig. 8). All but one simulation contribute to sea level, ranging from 1 mm sea level fall up to a 40 mm sea level contribution, and a mean sea level contribution of 21 mm. The Amundsen Sea Embayment sector (9) contributes to sea level rise in all but the two CNRM-CM6-1 forced projections. We note that for both sectors (4 and 9), sea level fall reflects mass loss there in the control, as all simulations contribute to sea level when the control simulation is not subtracted. For four



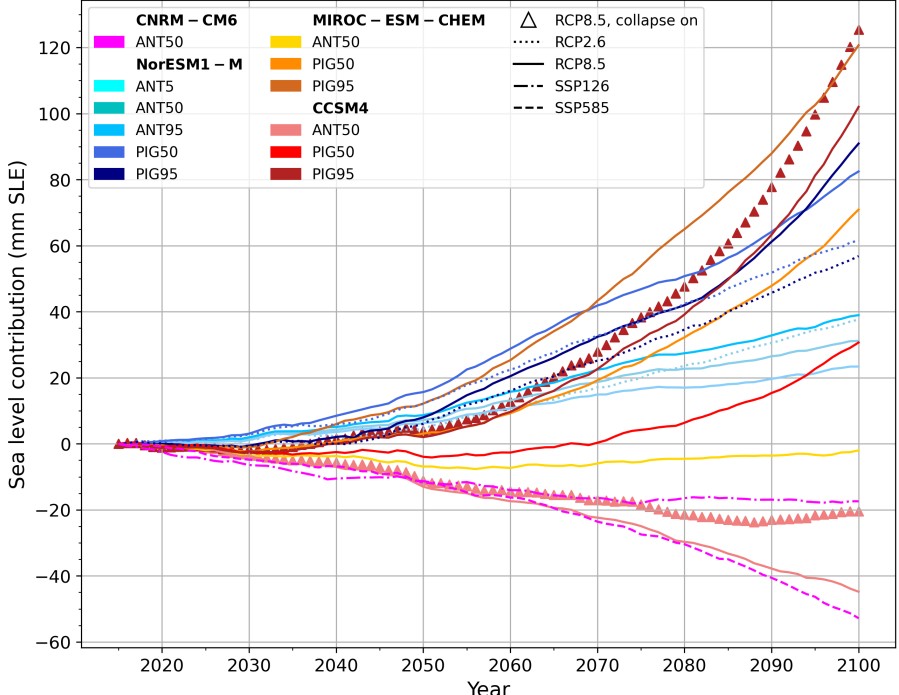

**Figure 4.** Sea level contribution (in mm) for all projections, relative to the control simulation, from 2015 to 2100.

experiments, the ASE is the sector with the largest projected contribution to sea level rise. Sea level contributions in the ASE range from 5 mm sea level fall to 36 mm sea level rise, with a mean sea level contribution of 12 mm.

In the Filchner-Ronne sector (14), nine simulations increase their VAF, with one simulation contributing -13 mm SLE – the largest mass gain in any sector and experiment. The Filchner-Ronne has a large area over which to accumulate mass. However, for two simulations with highest basal melt sensitivity under CCSM4 RCP8.5, the Filchner-Ronne sector undergoes losses equivalent to a 46 mm sea level rise. This is the largest contribution to sea level rise of any sector and gives Filchner-Ronne the largest projected range. The projected range in this sector illustrates the competing processes of increased accumulation under

warming on the one hand (Payne et al., 2021) and increased mass loss due to basal melting on the other. When sensitivity to ocean melt is low, increased accumulation dominates ocean melt-driven mass loss. Conversely, under higher ocean melt sensitivity, ocean melt-driven mass loss counteracts the warming-driven negative surface mass balance (SMB) feedback. Under the highest basal melt sensitivity, the loss of VAF in the Filchner-Ronne sector is 55 mm greater than under equivalent lower ocean sensitivity scenarios with the same forcing.

The other sector with a major ice shelf, the Ross Sea sector (7), contributes to sea level in eleven simulations. It has mass gain in some simulations, up to -11 mm SLE sea level contribution for lower ocean melt sensitivity experiments. The largest projected sea level contribution in this sector is 28 mm under NorESM1-M RCP8.5, with the second highest basal melt sensitivity.



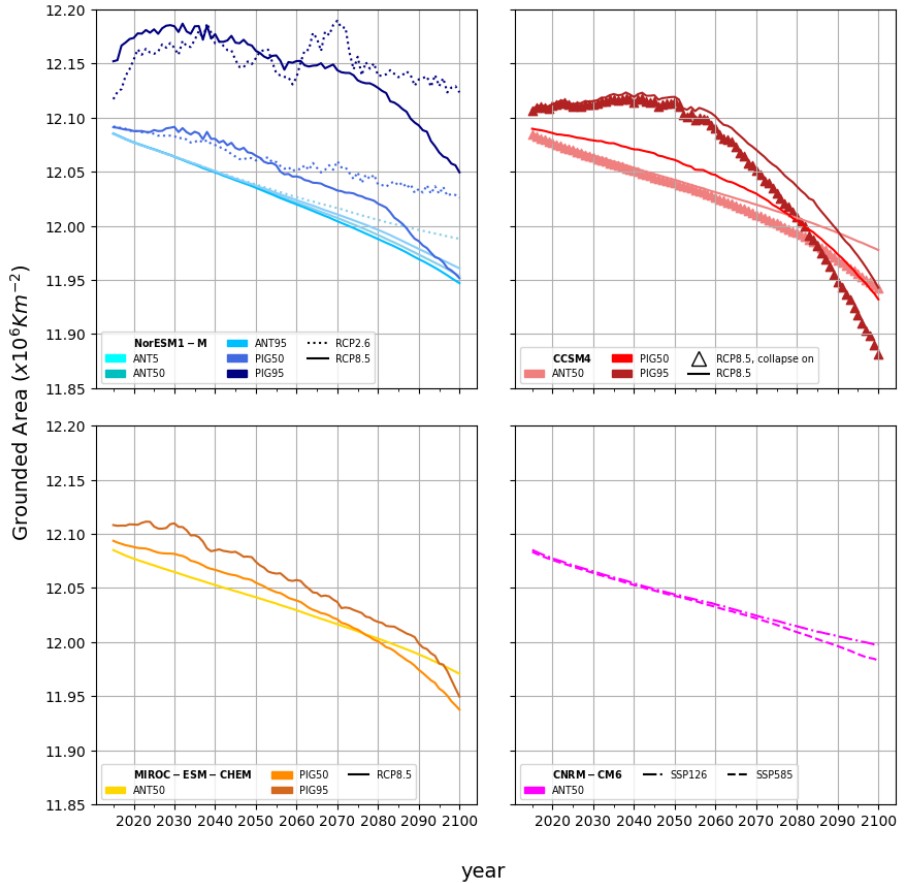

**Figure 5.** Grounded ice sheet area for all simulations from 2015 to 2100.

The second largest projected sea level contribution from the Ross sector is 24 mm for the same two experiments (CCSM4, RCP8.5, highest basal melt sensitivities) that have a 46 mm sea level contribution from the Filchner-Ronne sector.

For the ASE, the control simulation contributes 30 mm to sea level. Whilst subtracting the control can account for model drift, it may also in this instance be removing the sea level signal from ASE's long timescale to retreat initiated before 2015. Evidence for marine ice sheet instability in the ASE is equivocal, with the IPCC AR6 stating that observed flow regimes in the ASE are compatible with but not incontrovertible evidence of MISI (Fox-Kemper et al., 2021). In contrast, both the Ross Sea and Filchner-Ronne sectors steadily increase in VAF throughout the control simulation.

### 225 3.5 Patterns of thickness change

Across all simulations, the Thwaites and Pine Island catchments undergo thinning, as do the Totten, Queen Mary Land and George V Land glaciers (Fig. 9). The Ross and Filchner-Ronne ice shelves thin in the majority of simulations, with the





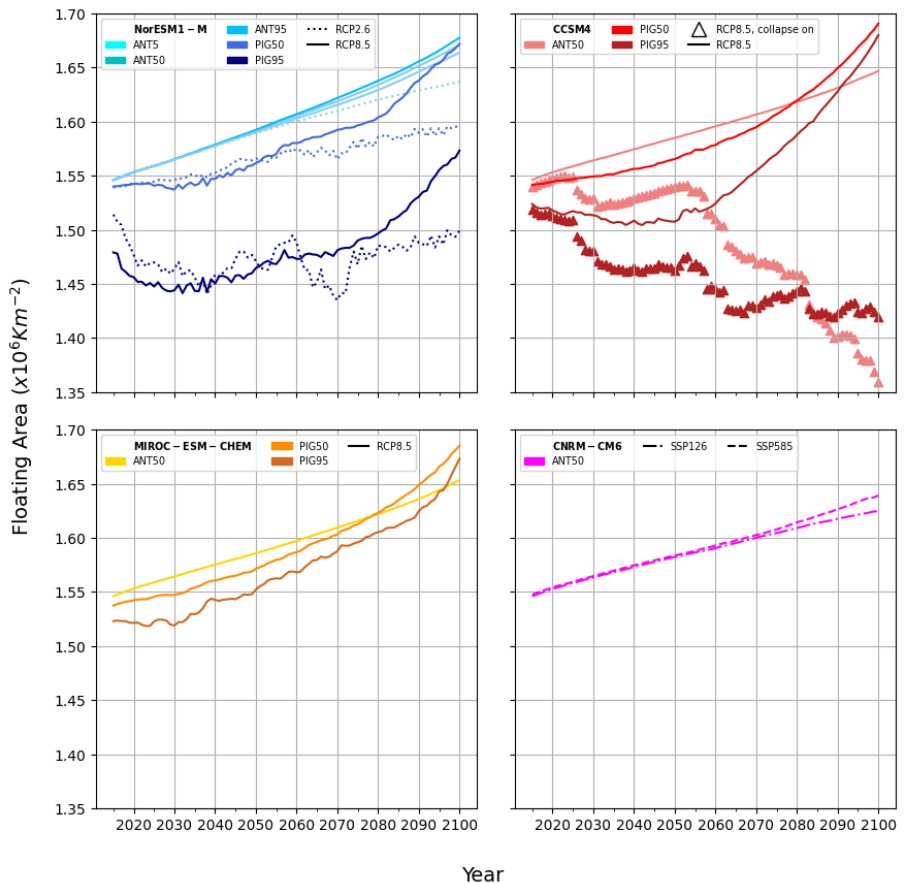

**Figure 6.** Floating ice sheet area for all simulations from 2015 to 2100.

exception of NorESM1-M RCP2.6 $PIGL$ simulations. Similarly, the Larson, Amery, Shackleton and Dronning Maud Land ice shelves thin in the majority of simulations (NorESM1-M RCP2.6 $PIGL$ simulations again excepted).

NorESM1-M RCP2.6 $PIGL$ simulations undergo thickening along the outer edge of the Ronne ice shelf, the majority of the Ross ice shelf, the Larsen ice shelf and those along the Weddell sector of Dronning-Maud Land (i.e. Brunt sector ice shelves)(Fig. 9 **k** and **l**).

## 4 Discussion

The results are discussed by the dependence on each modelling uncertainty or choice: GCM, emissions scenario, ice shelf
collapse and basal melt sensitivity. The last sections compare the BISICLES projections with those from other models in ISMIP6, and summarise the contribution of these projections to the synthesis by Edwards et al. (2021).



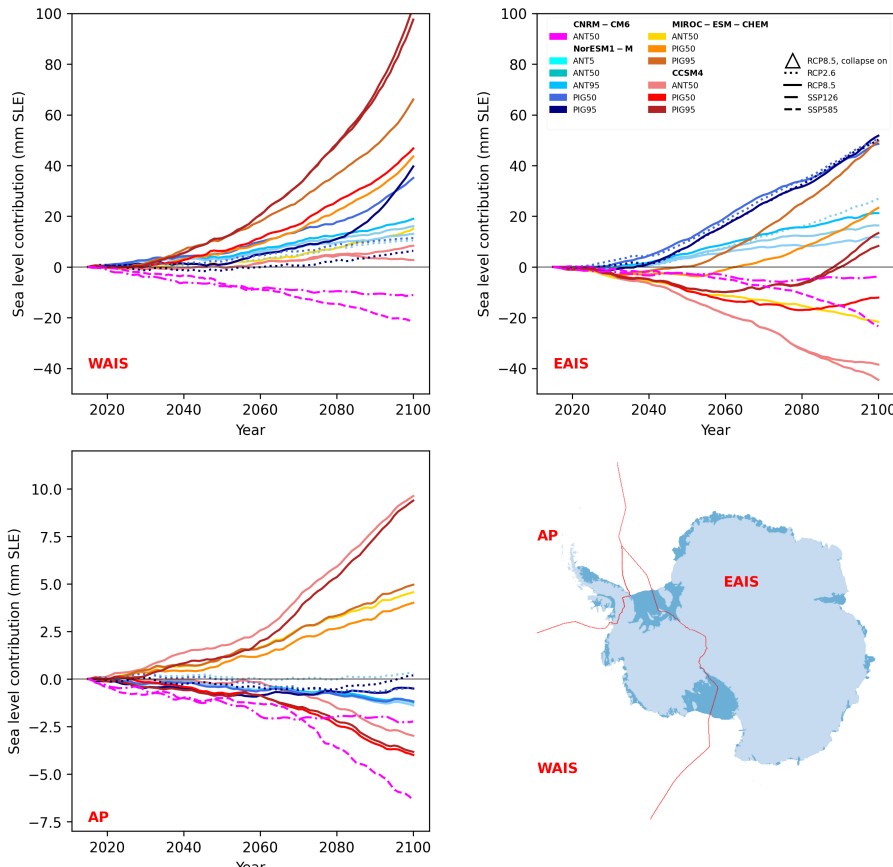

**Figure 7.** Sea level contribution (loss of volume above floatation) (in mm SLE) for the East Antarctic ice sheet (EAIS), West Antarctic ice sheet (WAIS) and Antarctic Peninsula (AP) from 2015 to 2100. Inset plot shows mask boundaries used to calculate regional change in volume above floatation

## 4.1 Dependence on GCM forcing

GCM-dependence of the projections is driven by differences in the magnitude and distribution of ocean thermal forcing, driving basal melt (Fig. 11), and the magnitude and distribution of surface mass balance over the ice sheet (Fig. 10).

To explore this, we can compare simulations with the same ice shelf basal melt sensitivity under the same emissions scenario. Under the $MeanAnt_{50}$ tuning and RCP8.5, the NorESM1-M forced simulation contributes 31 mm to sea level, MIROC-ESM drives a mass gain of 2 mm SLE VAF and CCSM4 drives a mass gain 45 mm SLE VAF. The GCM-dependence for these experiments arises from both the surface mass balance and ocean conditions. The CCSM4 RCP8.5 surface mass balance is positive over large regions of the ice sheet (Fig. 10), and MIROC-ESM is largely positive over the EAIS and interior WAIS

(Fig. 10). Conversely, NorESM1-M surface mass balance is negative over much of WAIS and the margins of EAIS (Fig. 10). Mass gain in EAIS compensates WAIS mass loss, driving an overall sea level fall for CCSM4 and MIROC-ESM forced



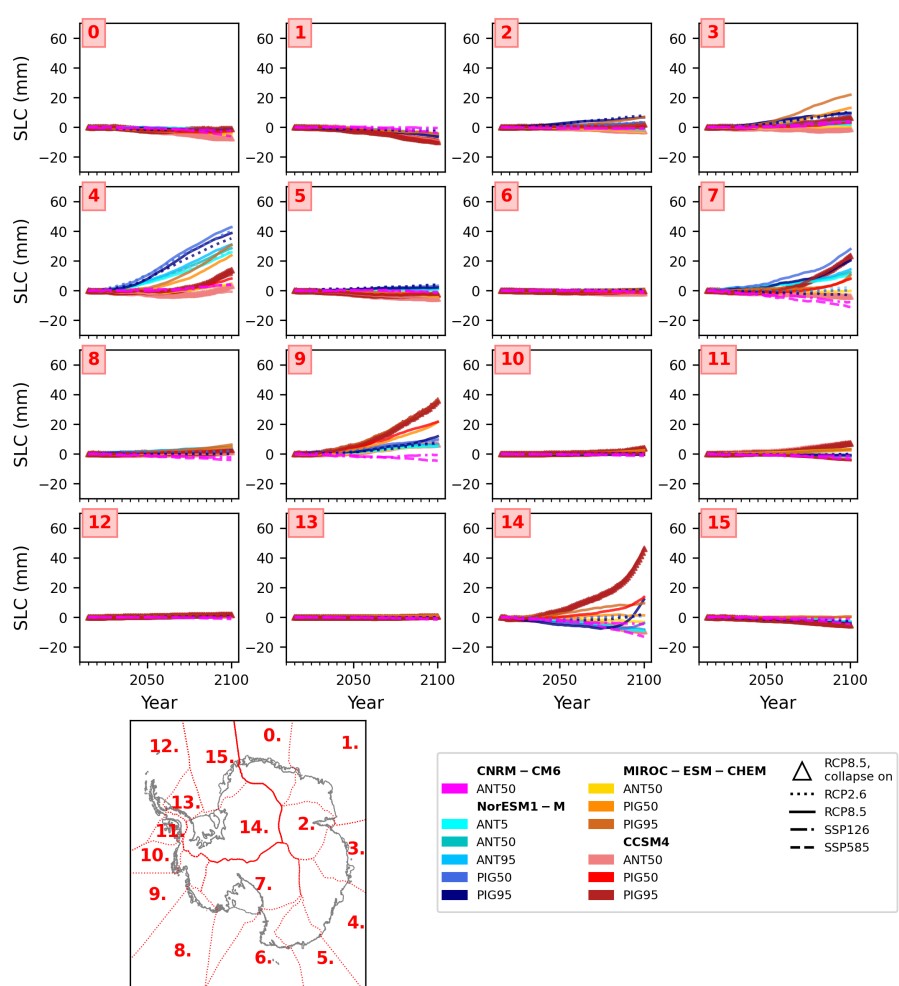

**Figure 8.** Sea level contribution by sector for all simulations. Basins are numbered are as follows: 0: Dronning Maud Land; 1: Enderby Land; 2: Lambert Glacier catchment; 3: Wilhelm II land; 4: Totten Sector; 5: George V Land; 6: Oates Land; 7: Ross Ice Shelf; 8: Getz ice shelf sector; 9: Amundsen Sea Embayment sector; 10: Abbott ice shelf sector; 11: George VI ice shelf sector; 12: Larsen sector; 13: Palmer Land; 14: Filchner-Ronne sector; 15: Brunt ice shelf sector.



**Figure 9.** Thickness change at 2100 relative to 2015 for all experiments, ordered by sea level contribution. Pink dashed line shows sea level contribution from 2015 until 2100 to give an indication of rate and magnitude. Black bold test corresponds to experiment number in Table 2



simulations under $MeanAnt_{50}$ (Fig. 7: pink (CCSM4) and yellow (MIROC-ESM) solid lines). Conversely, the NorESM1-M forced simulation loses mass from EAIS as well as WAIS (Fig. 7: blue solid line). This reflects the smaller response in the NorESM1-M atmosphere to RCP8.5 warming compared with other CMIP5 models (Barthel et al., 2020), which limits the

extent to which warming-driven increases in surface mass balance compensate ocean-driven losses. Under NorESM1-M, larger mass loss from EAIS compared with other GCMs is largely driven by grounding line retreat and greater loss of VAF in the Totten glacier catchment (Fig. 8, sector 4).

When higher basal melt sensitivity ($PIGL_{50}$) is used under the same emissions scenario, NorESM1-M again has the largest sea level contribution at 82 mm SLE. The MIROC-ESM forced simulation increases its sea level contribution to 71 mm SLE.

The CCSM4 forced simulation also undergoes mass loss, with a sea level contribution of 31 mm - compared with -45 mm SLE for $MeanAnt_{50}$. Under the highest basal melt sensitivity ($PIGL_{95}$), MIROC-ESM drives the largest mass loss - a sea level contribution of 121 mm. The next largest sea level contribution is from the CCSM4 forced simulation, with a sea level contribution of 102 mm SLE, with NorESM1-M driving the smallest sea level rise at 91 mm SLE. With the same surface mass balance forcing for each GCM but increased ice shelf basal melt sensitivity, model dependence becomes more influenced by

differences in ocean thermal forcing. Increased sea level contribution for MIROC-ESM is partly driven by increases in EAIS mass loss (e.g sectors 3 (Queen Mary Land) and 4 (Totten sector) in Fig. 8) where thermal forcing is high (Fig. 1). Both CCSM4 and MIROC-ESM forced simulations have large ASE sea level contribution under this higher basal melt sensitivity (sector 9 in Fig. 8), whilst NorESM1-M has lower thermal forcing and undergoes a smaller increase in sea level contribution with higher $\gamma_0$ in this sector.

For high basal melt sensitivity ($PIGL_{50}$ and $PIGL_{95}$) runs forced with the MIROC-ESM model, high basal melt is simulated under the Shackleton ice shelf in East Antarctica and drives higher sea level contribution for this sector compared with lower $\gamma_0$ simulations.

However, the contribution does not scale directly with shelf melt. This reflects the relatively limited buttressing effect of this ice shelf. It illustrates the way that unconstrained ice shelves can undergo significant melt with a limited impact on sea level

contribution (Fürst et al., 2016). Moreover, it illustrates how GCM-dependence is partially dependent on $\gamma_0$.

We also ran two simulations forced with a newer climate model (CMIP6) under newer emissions scenarios (SSPs). CNRM-CM6-1 has an equilibrium climate sensitivity (ECS) of 4.8°C (Meehl et al., 2020), similar to MIROC-ESM-CHEM (ECS = 4.7°C) the highest ECS CMIP5 model sampled in ISMIP6 and discussed in Payne et al. (Payne et al. 2021), but higher than the remaining CMIP5 models which have ECS of 2.9°C (CCSM4) and 2.9°C (NorESM1-M)(Flato et al., 2013). This drove large

positive surface mass balance in CNRM-CM6-1, leading to substantial accumulation (Fig. 2), offsetting dynamical losses from ocean melt-driven retreat. In many generally high mass loss sectors, such as the ASE and Totten catchments, CNRM-CM6-1 ocean thermal forcing is lower than other models, limiting ocean-driven mass loss under $MeanAnt_{50}$, and overall Antarctica contributes sea level fall under both scenarios (SSP1-2.6 and SSP5-8.5) for this model. It should, however, be noted that we only sample the $MeanAnt$ basal melt contribution, and could expect a larger sea level contribution for CNRM-CM6-1 with

greater melt sensitivity to thermal forcing at the base of the ice shelves.





## 4.2 Dependence on emissions scenario

**Figure 10.** Cumulative surface mass balance between 2015 and 2100 for all projections, ordered by sea level contribution. Dashed line is the sea level contribution through time for each run to give an indication of rate and magnitude.



The higher warming simulations (RCP8.5 for CMIP5 models and SSP5-8.5 for CMIP6) generally have higher surface mass balance over the continent (Fig. 10), consistent with larger precipitation flux under warming (Payne et al., 2021; Palerme et al., 2017; Frieler et al., 2015). The scenario dependence was then modulated by the value used for basal melt sensitivity.

Scenario-dependence was assessed for the two GCMs used to make projections under the low emissions scenarios (RCP2.6/SSP1-2.6): NorESM1-M from CMIP5 and CNRM-CM6-1 for CMIP6, calibrated to mean Antarctic melt rates. For the NorESM1-M simulations, the low emissions scenario leads to greater sea level contribution by 2100, i.e. counter to the intention of mitigating climate impacts: 38 mm under RCP2.6, compared with 31 mm under RCP8.5. This varies regionally: WAIS sea level contribution, for example, is smaller under RCP2.6 than RCP8.5 (10 mm vs 16 mm), as basal melting under RCP8.5 is greater

(Fig. 11 **h** vs **i**). Over the Ross shelf and WAIS, SMB is more negative under the higher emissions scenario (Fig. 10 **h** vs **i**). These factors together drive the higher mass loss in WAIS loss under RCP8.5 compared with RCP2.6 - consistent with other ISMIP6 ice sheet models forced by NorESM1-M, where mass loss is greater under RCP8.5 than RCP2.6 (Fig. 4(a) in Edwards et al. (2021)). For the EAIS and the majority of the Peninsula, the SMB scenario-dependence is reversed: SMB is higher under RCP8.5 compared with RCP2.6. This drives a smaller net sea level contribution in EAIS under RCP8.5 (16 mm vs 27 mm),

and a net mass gain in the Peninsula compared with RCP2.6 (1 mm SLE vs 0 mm SLE), which is consistent with most other ISMIP6 models (Fig. 4(c, d) in Edwards et al. (2021)).

In contrast to CMIP5, simulations forced by the CMIP6 model CNRM-CM6-1 project net mass gain under both emissions scenarios: sea level contributions are -53 mm under SSP5-8.5 and -17 mm SLE under SSP1-2.6. WAIS, EAIS and AP all have larger volume increase under SSP5-8.5 compared with SSP1-2.6. Unlike NorESM1-M, CNRM-CM6-1 consistently has higher

SMB under the higher emissions scenario across the majority of the ice sheet (Fig. 10 **a** vs **c**). Basal melt is higher under the higher emissions scenario for CNRM-CM6-1 (Fig. 11 **a** vs **c**), though not by enough to counteract the SMB increases, so VAF increases for all sectors (WAIS: 22 mm vs 11 mm SLE, EAIS: 23 mm vs 4 mm SLE, AP: 6 mm vs 2 mm SLE). This is consistent with other ISMIP6 projections forced with this climate model, where accumulation under higher emissions dominates over ocean melt-driven mass loss (Fig. 4 in Edwards et al. (2021)).

Two additional simulations beyond the ISMIP6 protocol (T71 and T73) were run to provide insight into the modulation of scenario-dependence by basal melt sensitivity. These apply NorESM1-M thermal forcing under RCP2.6 with $PIGL_{50}$ and $PIGL_{95}$ basal melt sensitivity parameters. For the median Pine island calibration ($PIGL_{50}$), high emissions lead to greater sea level contribution: 82 mm for RCP8.5 (experiment 13) compared with 62 mm for RCP2.6 (experiment T71). This again varies regionally. WAIS mass loss follows the overall scenario-dependence, with a larger regional sea level contribution under

RCP8.5 compared with RCP2.6 (35 mm vs 11 mm). EAIS losses again show the opposite pattern, with slightly larger sea level contribution under RCP2.6 than RCP8.5 (51 mm vs 49 mm). The Peninsula gains mass under both scenarios, with similar change in VAF for both scenarios (1 mm SLE).

For $PIGL_{95}$, high emissions also lead to greater mass loss and a larger sea level contribution: 91 mm SLE for RCP8.5 (experiment D52), compared with 57 mm SLE for RCP2.6 (experiment T73). The regional scenario dependence differs from

the $PIGL_{50}$ simulations. This time, both WAIS and EAIS losses follow the overall pattern of larger sea level contribution





**Figure 11.** Cumulative basal mass balance flux for ice shelves between 2015 and 2100 for all simulations, ordered by sea level contribution. Note that the color scale is inverted, so negative values indicate thinning. MIROC forced runs have large cumulative thinning of the Shackleton ice shelf. Dashed line is the sea level contribution through time for each run to give an indication of rate and magnitude.



under RCP8.5 compared with RCP2.6 (WAIS: 40 mm vs 7 mm; EAIS: 52 mm vs 50 mm). The Peninsula shows opposite sign contributions: the region loses mass under RCP2.6, but gains mass under RCP8.5.

These experiments informed the assessment of potential interactions between scenario and basal melt sensitivity (see Contributions to Edwards et al. (2021) below).

**4.3  Dependence on ice shelf collapse**

Two pairs of simulations explore the impact of shelf collapse on sea level contribution. All are forced with the CCSM4 climate model under RCP8.5. The first pair have ice shelf collapse on and off, with the $MeanAnt_{50}$ basal melt parameter value (experiment 12 and 8, respectively). The second pair is the same but with the $PIGL_{95}$ parameter value (experiment TD58 and D58), to explore the interactions between the basal melt parameter and shelf collapse. Experiment TD58 was beyond the

ISMIP6 protocol, and was performed to inform the synthesis by Edwards et al. (2021)

Including shelf collapse increases Antarctic sea level contribution by 25 mm SLE relative to 'no collapse' in both pairs of experiments (by region, the increase is: Peninsula: 13 mm; EAIS: 5-6 mm; WAIS 4-6 mm). However, the no collapse baseline is very different in the two basal melt parameterisations: for the $PIGL_{95}$ experiments, including shelf collapse increases the net mass loss; for the $MeanAnt_{50}$ experiments, it decreases the net mass gain. These two sets of projections informed the

assessment of interactions between ice shelf collapse and basal melt sensitivity (see Contributions to Edwards et al. (2021) below).

**4.4  Dependence on basal melt sensitivity**

To understand dependence of the projections on the basal melt parameter, experiments with the same GCM forcing and different $\gamma_0$ can be compared. Here all simulations have ice shelf collapse off. The most comprehensively sampled combination of GCM and scenario is NorESM1-M under RCP8.5: simulations were carried out for five basal melt sensitivity values, $MeanAnt_5$,

$MeanAnt_{50}$, $MeanAnt_{95}$, $PIGL_{50}$ and $PIGL_{95}$. Three of these values ($MeanAnt_{50}$, $PIGL_{50}$ and $PIGL_{95}$), which span most of the range, were carried out for NorESM1-M under RCP2.6, and for MIROC-ESM and CCSM4 under RCP8.5.

The overall $\gamma_0$-dependence for the majority of GCMs is one of increased sea level contribution under higher $\gamma_0$, as discussed throughout the results, though the nature of this relationship varies by region and model (Fig. 12). The Antarctic Peninsula is

fairly insensitive to increases in $\gamma_0$ (Fig. 12). In comparison with other ISMIP6 models, BISICLES has intermediate sensitivity to $\gamma_0$ (see Extended Data Fig. 6 in Edwards et al. 2021).

In the NorESM1-M experiments, increasing $\gamma_0$ from $PIGL_{50}$ to $PIGL_{95}$ leads to a more complex response in WAIS and EAIS than the simple increase in sea level contribution seen for other GCMs (Fig. 12). Under RCP2.6, the $PIGL_{95}$ simulation counterintuitively undergoes a smaller loss of VAF than the $PIGL_{50}$ simulation (Fig. 4: darker blue dashed lines).

Whilst localised thickening occurs intermittently for all GCMs and scenarios under $PIGL$ basal melt tuning (not shown), for NorESM1-M, thickening is pervasive enough to alter the dependence of net mass loss on $\gamma_0$. As can be seen in Fig. 9 subplots **k** and **l**, the Ross ice shelf thickens in both simulations, with more thickening under $PIGL_{95}$. Under RCP8.5, the $PIGL_{95}$



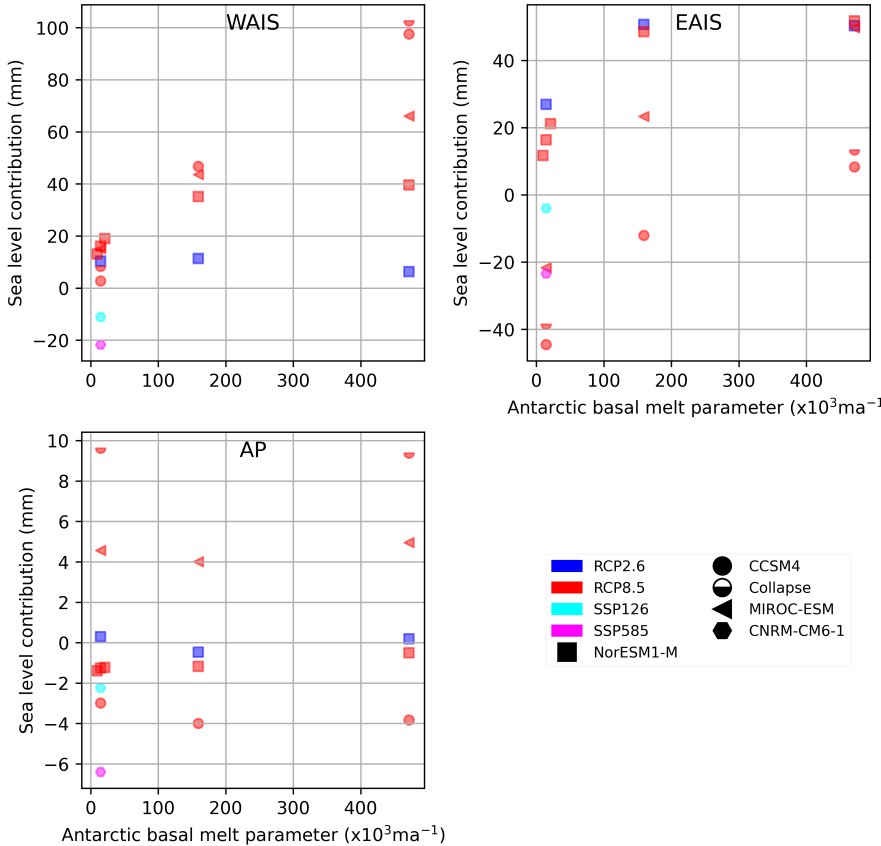

**Figure 12.** Sea level contribution from 2015 to 2100 relative to control for all simulations as a function of basal melt sensitivity ($\gamma_0$), shown for East Antarctic (EAIS), West Antarctic (WAIS) and Peninsula (PEN) ice sheets.

simulation also projects smaller contribution to sea level than $PIGL_{50}$ for most of the century, until overtaking in 2094 (Fig. 4: blue solid lines) for the whole ice sheet.

Previous studies using the same basal melt parameterisation have also noted ice shelf thickening as a result of refreezing under high basal melt sensitivity (Lowry et al., 2021; Lipscomb et al., 2021). Ice shelf refreezing under low thermal forcing is plausible, and present in observations and model simulations of Antarctic ice shelf cavities (Naughten et al., 2018; Adusumilli et al., 2020; Reese et al., 2018; Stevens et al., 2020). However, Lipscomb et al. (2021) modify the second term in equation 1 to avoid what they suggest is spurious melting and refreezing where sector-averaged thermal forcing plus the basin correction

($\delta T_{sector}$) is negative, by adding a limit such that:



$$
\begin{aligned}
m(x,y) = \gamma_0 \times & \left( \frac{\rho_{sw} C_{pw}}{\rho_i L_f} \right)^2 \\
\times & \left( TF(x,y,z_{draft}) + \delta T_{sector} \right) \\
\times & \, max \left( \langle TF \rangle_{draft \in sector} + \delta T_{sector}, 0 \right),
\end{aligned} \tag{2}
$$

This avoids negative values of m(x,y), which drive ice shelf thickening where $\langle TF \rangle_{draft \in sector} + \delta T_{sector}$ is negative. An earlier study exploring Antarctic sensitivity to future climate and model parameters used an alternative basal melt approach that also avoids refreezing of ice shelves by design (Bulthuis et al., 2019).

360   Our BISICLES version uses the ISMIP6 non-local melt parameterisation without modification (Jourdain et al., 2020). However, thickening of ice shelves as a result of the basal melt parameterisation is not permitted in this BISICLES B configuration. Thickening of ice shelves under the highest $\gamma_0$ values could therefore be a manifestation of tributary glaciers responding to strong ice shelf thinning and removal of buttressing, and advection of ice to grounding lines as ice streams speed up. Beyond 100 year time scales, initial thickening could therefore precede a larger long term sea level response. Future work could explore
365   whether melt sensitivity dependence for highest $\gamma_0$ values reverts to that seen for lower values (higher $\gamma_0$, more mass loss) over longer time scales.

  The Ross sector provides an example of an ice shelf and grounding line dynamic under $PIGL \, \gamma_0$ tuning that runs counter to our expectation: that higher $\gamma_0$ will increase shelf thinning, and enhance grounding line retreat. For this and other sectors under NorESM1-M RCP2.6 and RCP8.5 (e.g. Sector 4: Totten, and Sector 5: George V), sea level rise contribution under the
370   highest basal melt sensitivity ($PIGL_{95}$) is lower than under the second highest ($PIGL_{50}$) basal melt sensitivity (Fig. 8: blue solid lines). Figure 13 shows a transect through the grounding line at the terminus of Whillans and Mercer ice streams for $PIGL_{95}$ NorESM1-M RCP8.5 and $PIGL_{50}$ at three successive time slices (2015, 2050 and 2100). Also shown are the basin average thermal forcing for NorESM1-M RCP8.5. In the Ross Sea Sector, the grounding line under $PIGL_{95}$ is seaward of the equivalent $PIGL_{50}$ simulation grounding line for the duration of the simulation at the Whillans and Mercer ice streams
375   grounding line (Fig. 13). Ross sector ice streams drain around 40% of the West Antarctic ice sheet (Price et al., 2001), so changes to ice stream configuration along the Siple coast impact sea level contribution in the sector.

## 4.5   Comparison with other models

BISICLES is compared with other ISMIP6 models in Fig. 14, 15 and 16 for EAIS, WAIS and the Peninsula respectively. Whilst BISICLES generally lies within the range of other ISMIP6 models for WAIS and the Peninsula, for EAIS it shows a
380   systematically different response in some experiments.

  For EAIS (Fig. 14), BISICLES has the largest sea level contribution under mean Antarctic $\gamma_0$ tuning for NorESM1-M RCP8.5 forced simulations (Fig. 14). With the largest EAIS contribution in these experiments sourced from the Totten Glacier, this could suggest that BISICLES 1 km grid resolution at the Totten grounding line is resolving retreat not captured in lower



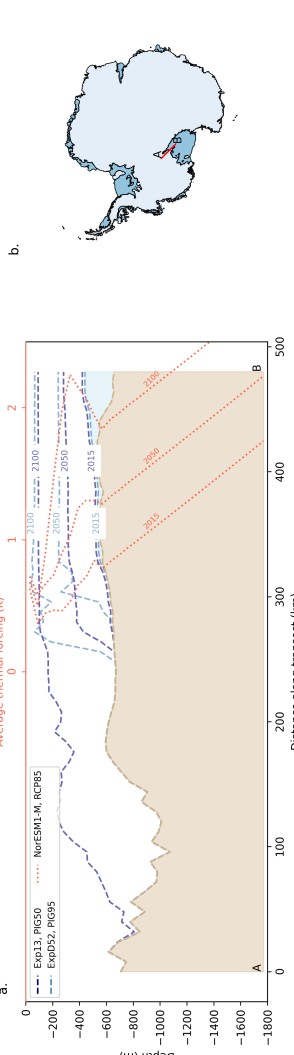

**Figure 13.** Siple coast transect for $PIGL_{50}$ and $PIGL_{95}$ experiments under NorESM1-M RCP8.5. Blue dashed lines show ice sheet base for years indicated. Red dashed lines show average thermal forcing with depth at successive time steps. The higher basal melt sensitivity ($\gamma_0$) run undergoes more thinning in outer shelf shown in transect, but grounding line retreats further inland for lower $\gamma_0$ run - though the shelf remains thicker for the latter.



resolution models (4-20 km for fixed resolution models; minimum 2 km for variable resolution models) - though we note that
Totten glacier can retreat at lower resolution (< 8 km) in BISICLES (Cornford et al., 2016).

For WAIS (Fig. 15), BISICLES projections for the core experiments tend to be mid-range and similar to two models with structural similarities: CISM, which is the other L1L2 physics model (though run on a fixed 4km gird), and UCI JPL ISSM, which also uses a variable mesh resolution. CISM additionally implements a sub-grid interpolation scheme to represent basal melt in partially floating cells (Lipscomb et al., 2021), which could account for its slightly larger sea level contribution under
NorESM1-M RCP8.5 core experiments for WAIS (Seroussi and Morlighem, 2018). Under increased basal melt sensitivity ($\gamma_0$), the CISM WAIS contribution is larger still. UCI JPL ISSM uses a variable mesh with finest resolution of 3 km near the margins, and has higher order physics (Seroussi et al., 2020). Agreement between ISSM and BISICLES for core WAIS simulations could reflect high resolution in both models, compared with other ISMIP6 models.We note that in the Marine Ice Sheet Model Intercomparison Project (MISMIP+), model physics had a less significant impact on simulated dynamics than basal sliding
law, which is based on Weertman sliding for both BISICLES and UCI JPL ISSM, at comparable resolution (Cornford et al., 2020). It is therefore less likely that agreement between the two models reflects consistency between BISICLES L1L2 and UCI JPL ISSMs higher order physics.

Overall, the SICOPOLIS model projects a larger sea level contribution compared with BISICLES in the majority of experiments, whilst GRISLI consistently projects a smaller sea level contribution. As noted in Edwards et al. (2021), SICOPOLIS
shows high sensitivity to ice shelf melt, likely due to its use of a floating condition for sub shelf melt - where basal melting is applied across the entire grid cell if the midpoint is at floatation. We note that MALI also uses a floating condition at the grounding line, and has a large WAIS contribution in core experiments. Conversely, GRISLI shows low sensitivity, which is ascribed to topographical biases in the initial condition making the model less sensitive to ocean-driven changes (Quiquet and Dumas, 2021).

**4.6  Contributions to Edwards et al. (2021)**

All simulations presented here were included in the synthesis of projections of global land ice contribution to 2100 sea level by Edwards et al. (2021), extending the ISMIP6 ensemble by an additional model compared with Seroussi et al. (2020) and Payne et al. (2021). Experiments beyond the main ISMIP6 protocol were also conducted to provide further exploration of sensitivities and interactions.

As outlined in Section 4.3, the increase in sea level with collapse on is almost identical for both basal melt sensitivities sampled ($MeanAnt_{50}$ and $PIGL_{95}$). Along with results from the same experiments in ISSM, this is the basis for the conclusion in Edwards et al. (2021, section "Ice shelf collapse versus basal melt") that contribution due to ice shelf collapse does not significantly increase with higher values of $\gamma_0$.

A further finding that was supported with these projections is highlighted in the Section "Retreat and basal melt versus
temperature" of Edwards et al. (2021). Sampling $PIGL$ basal melt sensitivity under RCP2.6 (T71 and T73) to compare with RCP8.5 projections shows that the spread of projections is smaller under the former scenario. This result is confirmed in complementary experiments with ISSM, as presented in Edwards et al. (2021).





**Figure 14.** East Antarctic Ice Sheet (EAIS) sea level contribution (SLC) comparison with other ISMIP6 simulations from 2015 to 2100. Data from Edwards et al. (2021).





**Figure 15.** West Antarctic Ice Sheet (WAIS) sea level contribution (SLC) comparison with other ISMIP6 simulations from 2015 to 2100. Data from Edwards et al. (2021). BISICLES shown in black solid line.



**Figure 16.** Antarctic Peninsula (AP) sea level contribution comparison (SLC) with other ISMIP6 simulations from 2015 to 2100. Data from Edwards et al. (2021). BISICLES shown in black solid line.





## 4.7 Limitations

For NorESM1-M RCP2.6 $PIGL_{95}$, the sea level contribution at 2100 is lower than that projected under $PIGL_{50}$. However,
the trajectory of mass loss in Figure 4 indicates that $PIGL_{95}$ will overtake $PIGL_{50}$ beyond 2100. To confirm this, extending
these simulations beyond 2100 would be a worthwhile extension on this work. More broadly, IPCC AR6 extrapolates mass
trends from 2100, the end of the simulation period for the model inter-comparisons it draws on, to project sea level to 2150 - a
time horizon that is increasingly policy relevant for long-lived infrastructure (Fox-Kemper et al., 2021). With ice sheet model
simulations beyond 2100, longer-term sea level projections could be informed by physics-based models, without the need to
assume mass trends.

Another informative extension on the work presented here would be to more comprehensively explore model uncertainties.
We explored five of the six $\gamma_0$ values provided by ISMIP6, omitting an intermediate ($PIGL_5$) values from our experiments.
Future simulations could include this $\gamma_0$ value. Moreover, whilst we were limited to the discrete $\gamma_0$ values provided by ISMIP6,
as calculating intermediate values was beyond the scope of this work, it is in practice a continuous parameter. Similarly, we
did not explore the full range of boundary conditions provided by ISMIP6, or all possible combinations of uncertainties.
Future work could more systematically quantify uncertainties in GCM forcing, $\gamma_0$ values and parameter interactions in a
comprehensive ensemble design, such as a Latin Hypercube.

Ice sheet initial condition plays an important role in model uncertainty (Seroussi et al., 2019). However, exploring initial
condition uncertainty was beyond the scope of this study. Future work could explore how consistent the BISICLES response
to future climate and parameter uncertainty is, when the simulations begin from a different modern initial condition, such as
one based on BedMachine (Morlighem et al., 2020) or Bedmap3 (Frémand et al., 2023).

The impacts of solid earth changes on projected ice sheet contribution to sea level are not explored for ISMIP6 (Nowicki
et al., 2016), and we do not include them in our experiments. Some projection studies have incorporated simplified models of
ice sheet bedrock interactions (Coulon et al., 2021; DeConto and Pollard, 2016; DeConto et al., 2021; Bulthuis et al., 2019).
Bulthuis et al. (Bulthuis et al. 2019) find that the capacity of bedrock deformation to stabilise the ice sheet, by deforming as the
ice thins to maintain contact and slow un-grounding, is limited over the 21st century for the slow bedrock response times that
characterise most of Antarctica(Bulthuis et al., 2019). However, bedrock underlying West Antarctica, where mantle viscosity
is low, can deform more rapidly than elsewhere in the continent (Barletta et al., 2018). Rapid viscous deformation driven
bedrock processes, alongside elastic bedrock deformation (Larour et al., 2019), have the potential to stabilise the ice sheet on
sub-centennial time-scales - limiting sea level contribution (Kachuck et al., 2020). Conversely, bedrock uplift as marine ice
sheets retreat can reduce accommodation space for ocean water, and therefore increase GMSL (Pan et al., 2021; Yousefi et al.,
2022).

Future work could improve our modelling framework to capture fast-responding West Antarctic bedrock, as in Kachuck et
al. (2020). This could help quantify the role of bedrock deformation in slowing ice sheet mass loss, and give a more detailed
picture of future Antarctic sea level contribution.



## 5  Conclusions

We have presented projections for the Antarctic ice sheet this century performed with the BISICLES model for ISMIP6. The response to emissions scenario, i.e. global warming, is strongly modulated by basal melt sensitivity ($\gamma_0$). Under warm climates, if $\gamma_0$ is tuned to high melt rates (derived from Pine Island glacier) then strong basal melt drives dynamical loss and large sea

level contributions. However, if basal melt sensitivity is low, Antarctica tends to gain mass relative to the control simulation, due to increasing accumulation.

With a high equilibrium climate sensitivity of 4.8°C (Meehl et al., 2020) and relatively high surface mass balance (Fig. 2), projections forced by the CMIP6 global climate model CNRM-CM6-1 gained mass for both simulations presented here (high and low emissions). However, these both used low basal melt sensitivity values ($MeanAnt_{50}$); we would expect greater mass

loss and larger sea level contribution for higher values. The climate model CCSM4 also drives sea level fall under the high emissions scenario RCP8.5 with low basal melt sensitivity $MeanAnt_{50}$, due in part to its large surface mass balance, though with high basal melt sensitivity ($PIGL$) and ice shelf collapse on this climate model drove the largest sea level rise. This highlights the importance of constraining plausible values of basal melt sensitivity for Antarctica under future warming, as this moderates the balance between accumulation-driven sea level fall on the one hand, and ocean melt-driven dynamical loss on

the other. We note that recent studies find the $PIGL$ tuning of the ISMIP6 parameterisation leads to greater error, relative to an ocean model, in yearly integrated melt than $MeanAnt$ (Burgard et al., 2022).

However, increasing the basal melt sensitivity value did not always increase sea level contribution: the response varied under different scenarios, climate and ice sheet models, regions and time periods. This demonstrates a nonlinear dynamic response to large ocean melt perturbations. We expect that beyond 2100, larger $PIGL$ $\gamma_0$ values would drive consistently larger sea level

contribution under all scenarios.

Ice shelf collapse increased sea level contribution overall, and had a comparable effect on sea level contribution for both basal melt sensitivity values tested ($MeanAnt_{50}$ and $PIGL_{50}$).

*Code and data availability.*  Code to reproduce analysis and figures will be available on github https://github.com/jone006/imsip6_bisicles_paper in due course. BISICLES model code is available on https://anag-repo.lbl.gov/svn/BISICLES/public/branches/ISMIP6-AIS/code/. Data is

available on request, and will be publicly available in due course

*Author contributions.*  S.N lead the overall ISMIP6 project, and H.S. coordinated the Antarctic projections for ISMIP6. T.E developed additional experiments based on the ISMIP6 protocol, and formulated this study along with J.O.

D.M and C.S conducted core ISMIP6 experiments, J.O conducted non-core experiments and those outside the ISMIP6 protocol. C.S, D.M and S.C developed software for processing model outputs. J.O developed software to analyse and visualise all results presented in this paper.

S.C and D.M were lead developers of the BISICLES ice sheet model, and developed the model set-up used for these experiments. D.M



provided access to the National Energy Research Scientific Computing Center (NERSC) on which experiments were conducted, and storage access.

J.O wrote the first draft, all authors provided feedback and edits to improve the manuscript.

*Competing interests.* The authors declare that they have no competing interests

*Acknowledgements.* James O'Neill acknowledges support from the UK Natural Environment Research Council (NERC) (grant NE/L002485/1). Support for this work was provided through the Scientific Discovery through Advanced Computing (SciDAC) program funded by the U.S. Department of Energy (DOE), Office of Science, Biological and Environmental Research and Advanced Scientific Computing Research programs, as a part of the ProSPect SciDAC Partnership. Work at Berkeley Lab was supported by the Director, Office of Science, of the U.S. Department of Energy under Contract No. DE-AC02-05CH11231. This research used resources of the National Energy Research Scientific

Computing Center (NERSC), a U.S. Department of Energy Office of Science User Facility located at Lawrence Berkeley National Laboratory, operated under Contract No. DE-AC02-05CH11231 using NERSC award ASCR-ERCAPm1041.

We thank the Climate and Cryosphere (CliC) effort, which provided support for ISMIP6 through sponsoring of workshops, hosting the ISMIP6 website and wiki, and promoted ISMIP6. We acknowledge the World Climate Research Programme, which, through its Working Group on Coupled Modelling, coordinated and promoted CMIP5 and CMIP6. We thank the climate modeling groups for producing and

making available their model output, the Earth System Grid Federation (ESGF) for archiving the CMIP data and providing access, the University at Buffalo for ISMIP6 data distribution and upload, and the multiple funding agencies who support CMIP5 and CMIP6 and ESGF. We thank the ISMIP6 steering committee, the ISMIP6 model selection group and ISMIP6 dataset preparation group for their continuous engagement in defining ISMIP6. This is ISMIP6 contribution No X





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
