# Peer review of "ISMIP6-based Antarctic Projections to 2100: simulations with the BISICLES ice sheet model"

_EGUsphere, 2024_

## Referee Comment (RC1)

A review of "ISMIP6-based Antarctic Projections to 2100: simulations with the BISICLES ice sheet model" by O'Neill et al.

In this manuscript, the authors conduct numerical simulations of Antarctic Ice Sheet using BISICLES, and investigate the impacts of climate forcing and model configuration (e.g., ice shelf collapse) on the model sensitivities. Overall, I think this study is a necessary follow-up of Edwards et al. (2021) and Seroussi et al. (2020), and is within the scope of The Cryosphere. The authors conducted comprehensive and detailed experiments regarding the uncertainties of the contribution of Antarctic Ice Sheet to sea level rise. In principle, I endorse its publication after the following questions are properly addressed.

L1-6: please complete the affiliation informaiton

L32-33: I think it is probably better to say something differently between EAIS and WAIS if the SMB change pattern is different, which is in consistent with the experiment design of this paper.

L34: remove extra space before "10 mm"

L44: remove extra comma after "models"

L58: put the sentence "BISCILES ISMIP6 ..." in a different paragraph and elaborate the reason you choose BISCILES for this study.

L81: I think you should also add Seroussi et al. (2014) along with Cornford et al. (2016) to represent the sub-grid GL scheme.

L85-86: Regarding "m=1/3 and Coulomb friction coefficient", I think you still need put some basic important equations here, like several equations describing the L1L2 approximations, and then you can properly get those model parameters settled somewhere.

L91: Regarding "the calving front is fixed", but there are also several experiments that you calve all ice shelves away, correct?

L114: it should be "...found in Jourdain et al. (2020)"

L130: extra space after "collapse"

L134: extra space betweem mm and a-1

L135: So the ice shelves are removed in a sudden?

L154: wrong citation format

L157: Fig. 3(b) -> Fig. 3b

L160: Fig. 3(a) -> Fig. 3a

Figure 3: please use white background for both Fig 3a and b.

Table 2: Looking at the Collapse On experiments here, it reminds me ABUMIP. Have you compare your results with that of ABUMIP? If not, I suggest doing some analysis. Also, the numbers at the "Sea level contribution" column are not exactly the same as in your following figures (e.g., figure 11), please check.

Figure 5 and 6: for y label, the unit is km2

Figure 8: in the caption: Basins are numbered as follows...

L268-270: I don't think this discussion is necessary here, as SLR is directly contributed by VAF and there is very complex relationship between basal melt of ice shelf and SLR.

Figure 10: For SMB, why are all ice shelves are missing? In addition, the spatial pattern here is not clear. Maybe you should try another way to plot them - maybe log scale?

L290: you do not have to mark "h" and "i" in bold here

L292: Fig. 4a

L294: you mean (27 mm vs 16 mm)?

L296: Figs. 4c and d

L305-312: I think you should say something about the two major ice shelves in WAIS, Filchner Ronne and Ross ice shelf. For example, does the basal melt of Filchner Ronne ice shelf increase nearly proportionally to that of Ross ice shelf?

L325-330: So can we get a conclusion that the buttressing of ice shelf can contribute a 20-30 mm SLR?

Equation 2: you still need to elaborate the meaning of each term in this equation.

Section 4.5: **This is probably the major concern of this study**. The analysis and figures here seems to be a bit overlapping with previous studies like Seroussi et al. (2020). I doubt if it is necessary to compare BISICLES with all other different types of models. Maybe you can just compare it with other higher order models, which I think makes more sense. Then it might be possible that you can put all comparisons in a single figure, insteading of showing them similarly in 3 figures (14-16).

Figure 13: Please put the thermal forcing curves in a separate plot, which will make a clearer and nicer figure.

Section 4.7: It is not clear to me if you have done the hindcast experiment. Please clarify.

---

## Referee Comment (RC3)

**Review of O'Neill et al. 'ISMIP6-based Antarctic Projections to 2100: simulations with the BISICLES ice sheet model'**

**General comments**

This manuscript presents new experiments designed to make projections of the Antarctic ice sheet contribution to sea level rise by 2100. To do this the authors follow the ISMIP6 protocol using the BISICLES ice-sheet model and climate forcing from CMIP5 models. They also run additional simulations that were not part of the original ISMIP6 framework using forcing from a CMIP6 model. The key finding is the important compensating effect surface accumulation can have on ocean-driven ice loss under certain combinations of parameters. I enjoyed reading the paper and feel that it is a useful addition to the originally presented ISMIP6 experiments and those included in the synthesis of Edwards et al., 2021. I think it is appropriate for the journal but I have some recommend changes prior to publication. Most of these relate to improving the readability of the manuscript, and some of the more substantial ones are included below. Minor changes are included in the specific comments in the following section.

1. I found the both the Introduction and Discussion sections to be unnecessarily long. There are also some instances of duplicated text. I think the Introduction would benefit from being shortened and more concise in places.

2. I think the paper would benefit by presenting more detail on the initial ice sheet model state and comparing this to observations and the other model states included in the ISMIP6 experiments. It would be good for the Control simulation section of the Results to include more detail on this, and perhaps an additional figure (see Specific comments).

3. I have several comments with regards to the Figures. 1) I think the number of figures in the manuscript is a little excessive and I wonder if all of them are absolutely necessary for the main messages of the paper. I think some could be moved to a Supplementary Information document at the least. 2) In many cases the Figure captions are not as detailed as they need to be and the individual panels on almost all figures should be labelled. For both points I've included suggestions in the specific comments below.

4. The Discussion as mentioned above is very long and in places presents a stream of new results, and often without any reference to Figures/Tables where the results can be found making it difficult to follow. I think it would be good if each section of the Discussion is considered whether it belongs there or in the Results. At the very least I think Section 4.1 should be moved to the Results. In general it would be good to consider what detail is/is not necessary, and I've made some suggestions in the specific comments. Also the Discussion mostly compares the results to those of Edwards et al., 2021 and Seroussi et al., 2020, and could benefit from referencing additional papers to support your findings.

5. The Limitations section could benefit from adding several more. For example how might your results have been different if you had used a different basal melt parameterisation? Also it would be good to mention the choice of sliding law, and how this is a potential source of uncertainty that has not been explored in these experiments. Finally, it would be good to discuss some unaccounted for processes in your experiments, e.g. iceberg calving (fixed ice front), and the SMB-elevation feedback/evolving topography during the simulations (that you do mention on Line 128).

6. I think the Conclusions would benefit from being revised, making sure they are a brief summary of the paper and not introducing any new information, and towards the end providing summary sentences on the key finding(s) of the paper and the wider importance.

In addition to the points raised above I have a number of specific line comments that are included below, that hope to help improve the clarity in places.

**Specific comments**

**Line 6:** could include examples of the uncertain ice sheet processes you are exploring

**Line 8-10:** I think this sentence would benefit from being rephrased for clarity

**Line 12:** "increases sea level contribution by 25 mm" relative to what? I think the whole range of sea level contributions from your experiments should be stated in the abstract.

**Line 16:** Does this mean "dominated the total sea level change of 20 cm" or ice sheets contributed to 20 cm?

**Line 36:** "future" Antarctic contribution to sea level? Also "represent a greater range of interactions and dynamic processes" reads a bit awkwardly, could it be rephrased, or examples of these "interactions" added?

**Line 44-47:** These sentences duplicate the information in the previous paragraph (lines 39-43), I suggest combining and removing the second paragraph of the two.

**Line 49:** I think the estimate of sea level contribution from the ISMIP6 experiments could come earlier/in the first paragraph of the introduction.

**Line 60:** See later comments and in the general comments section. How does your initial model state here compare to the one included in the initMIP experiments?

**Line 64:** change to "identified in Barthel" to make it clear that these models were not created in this paper, but evaluated.

**Line 79:** So the model set-up in this paper was exactly the same as the one presented in the initMIP experiments? that is not totally clear from the text. Does that include the choice of basal sliding law, grid resolution, etc? Despite this, I think more space could be given here and in Section 2.3 on how this initial state compares to that of the other models and to observations.

**Line 82:** A figure of the model mesh would be good in a Supplementary Information document

**Line 85:** Perhaps state why this sliding was chosen.

**Line 91:** How long was this relaxation run? and how does the ice sheet state deviate from the one arrived at after the inversion? Some figures on the change in ice thickness/speed during this relaxation and the comparison to observations at the end of the relaxation/start of the control simulations would be good.

**Line 98:** Perhaps cite Goelzer et al., 2020 here https://doi.org/10.5194/tc-14-833-2020

**Line 100:** The first few sentences of this section duplicate the information already presented in the Introduction. I suggest removing either here or from the introduction (lines 63-66).

**Line 105:** Is this the same selection of CMIP6 models included in Payne et al. 2021, if so state this.

**Line 118:** Perhaps state here that the approach of averaging over the surface ocean is the same as used in ISMIP6 and add the reference.

**Line 125:** Why did you not use PIGL5th? State this in the text.

**Figure 1:** "(0-500m)"

**Line 127:** Why did you choose to use the surface mass balance from Arthern et al., 2006 rather than RACMO?

**Figure 2:** The units between the spatial plots and line graph in Figure 1 are the same. It would be good to do that for the SMB anomalies in Figure 2 as well.

**Line 141-142:** This sentence is almost identical to the one on line 127-128, remove one of them.

**Line 142:** As mentioned above, it would be good to have more details on your initial model state, the relaxation run, and the subsequent control simulation. It would be useful to include a figure of the surface mass balance and basal melt rates used during the control simulation.

**Line 147:** It's confusing to have two sub-sections in close succession using the same heading.

**Line 148:** I don't think it's useful to include the total mass here, remove it and just focus on volume above flotation (as done in the rest of the manuscript).

**Line 151:** None of the place names in this paragraph are labelled on a figure. I guess the assumption is people will know where they are, but it would be nice to label the main ice shelves.

**Line 158:** It's almost impossible to see the grounding line retreat on Fig. 3b, I suggest making the map much larger or even creating an inset(s) for the key regions talked about in this paragraph.

**Line 165:** Replace "majority" with the number of simulations that show net mass loss. Also put this range of sea level contribution in the abstract.

**Line 173:** State that higher values of $\gamma_0$ are darker colours in the figure caption.

**Line 187:** Be consistent with use of negative numbers for mass gain/sea level fall. Improve the readability of the following sentence.

**Line 194-197:** Am I correct in thinking you didn't merge these basins and this was done in Jourdain et al., 2020? If so, just state that you use the basins from Jourdain et al., 2020 and remove the detail here.

**Line 200:** It would be nice to see the perturbation experiments alongside the control simulation (without subtracting it). Perhaps a figure in a supplementary information document.

**Line 206:** Make it clear when you say "Filchner-Ronne has a large area..." you are talking about the catchment not just the ice shelf?

**Line 209:** State the range.

**Line 210:** The important compensating effect of accumulation on ocean-driven mass loss has been noted for the Filchner-Ronne region by a number of previous studies, some of which might be worth citing here e.g. Cornford et al. 2015 and Wright et al., 2014.

**Section 3.4:** Throughout this section there are almost no references to any figures where the results can be found.

**Figure 5:** This figure caption is incomplete. State what each panel shows and add the CMIP forcing to the caption as well.. Also add panel labels to the figure.

**Line 224:** Could state that an increase in VAF in the Filchner-Ronne and Ross regions in the control simulations is consistent with the present-day trend in VAF and some references e.g. Rignot et al., 2019.

**Line 225:** Given that this section of the results is quite short it could just be combined with the basin discussions in the previous section. Figure 9 could probably be removed or moved to a supplement.

**Line 222:** Be careful here, this sentence is quite confusing, you are saying that it is "equivocal" that there is potential for MISI in the ASE, but that there is not evidence that for MISI happening yet. I would rephrase this to improve the clarity.

**Figure 7:** Needs panel labels and increase the font size in the legend.

**Section 4.1.** The entirety of this section reads as a stream of new results and reference to new figures. I think this would be better moved to the results. I also don't know how useful the spatial discussion is, and was wondering if Figures 10/11 would be better in a supplement and replaced with a figure that shows integrated ice sheet/regional surface mass balance and dynamic ice discharge instead.

**Figure 8:** The y-axis label is "SLC" but throughout the manuscript you use "SLE" I would be consistent with one or the other. Also I can see the reason for a consistent y-axis scale, but it means it some basins it is impossible to see any change through time, so I recommend modifying the y-axis scales individually.

**Figure 9:** I am not sure about showing the SLC lines ontop of the spatial plots, they are difficult to see and in several cases they obscure the text. Also the colourbar for the thinning is quite saturated. Could you just show the inland/grounded ice thinning only instead?

**Line 278:** Not sure this should be the start of a new paragraph. Also, is the same true for other unconstrained ice shelves, e.g. Thwaites?

**Line 279:** I did not find a clear statement as to why you did not also use the PIGL sensitivities for these CMIP6 model runs?

**Figure 10:** The caption is incomplete. Refer to experiments in Table 2 (as done in Fig. 9). See earlier suggestions to replace this figure with integrated ice sheet/region wide SMB.

**Line 296:** This section also reads as new results. If it remains as a Discussion section I suggest adding references to more papers that have shown similar results. E.g. for statements about compensating effects of increased precipitation with warming in East Antarctica mass for example Jordan et al., 2023- https://doi.org/10.1038/s41467-023-37553-2 and Stokes et al. 2022 and references therein https://doi.org/10.1038/s41586-022-04946-0).

**Line 305-312:** This paragraph again is a list of new results without any reference to any figures/tables where these results can be found. Same is true for the following paragraph.

**Figure 12:** panel labels

**Line 356:** Given that this equation is not used in this manuscript it feels a bit out of place in the Discussion, I would suggest removing it and just referring to the equation number in that paper instead.

**Line 414:** There are several references to Sections of Edwards et al., 2021. I think in most cases it would be better to briefly summarise the findings in that paper in your manuscript so the reader doesn't have to frequently go back and forth.

**Figures 14, 15, 16:** The text describing the details of each simulation overlaps large parts of the figure/results, I suggest removing and just directing the reader to Table 2, where the details of the experiments are.

**Line 421:** It is my understanding that future work is underway to extend simulations to 2300? is that worth mentioning here?

**Line 427:** Can you state why you omitted the PIGL5 values?

**Technical corrections**

**Line 16:** Remove ".0" from "20.0cm"

**Line 440:** Bulthuis reference error

---

## Author Comment (AC1)

Reviewer 3 feedback

We are grateful to Reviewer 3 for their detailed and constructive comments, and helpful suggestions. Where minor suggested edits have been proposed, these are incorporated into the revised paper and we do not include them below. We respond to more substantial comments in the following. Reviewer comments are bold and italicised for clarity, followed by our response.

***1. I found the both the Introduction and Discussion sections to be unnecessarily long. There are also some instances of duplicated text. I think the Introduction would benefit from being shortened and more concise in places.***

See response to comments below.

***2. I think the paper would benefit by presenting more detail on the initial ice sheet model state and comparing this to observations and the other model states included in the ISMIP6 experiments. It would be good for the Control simulation section of the Results to include more detail on this, and perhaps an additional figure (see Specific comments).***

See responses below.

***3. I have several comments with regards to the Figures. 1) I think the number of figures in the manuscript is a little excessive and I wonder if all of them are absolutely necessary for the main messages of the paper. I think some could be moved to a Supplementary Information document at the least. 2) In many cases the Figure captions are not as detailed as they need to be and the individual panels on almost all figures should be labelled. For both points I've included suggestions in the specific comments below.***

See responses below.

***4. The Discussion as mentioned above is very long and in places presents a stream of new results, and often without any reference to Figures/Tables where the results can be found making it difficult to follow. I think it would be good if each section of the Discussion is considered whether it belongs there or in the Results. At the very least I think Section 4.1 should be moved to the Results. In general it would be good to consider what detail is/is not necessary, and I've made some suggestions in the specific comments. Also the Discussion mostly compares the results to those of Edwards et al., 2021 and Seroussi et al., 2020, and could benefit from referencing additional papers to support your findings.***

See responses below.

5***. The Limitations section could benefit from adding several more. For example how might your results have been different if you had used a different basal melt***

*parameterisation? Also it would be good to mention the choice of sliding law, and how this is a potential source of uncertainty that has not been explored in these experiments. Finally, it would be good to discuss some unaccounted for processes in your experiments, e.g. iceberg calving (fixed ice front), and the SMB-elevation feedback/evolving topography during the simulations (that you do mention on Line 128).*

As suggested, we will expand the limitations section to include those suggested.

**6. I think the Conclusions would benefit from being revised, making sure they are a brief summary of the paper and not introducing any new information, and towards the end providing summary sentences on the key finding(s) of the paper and the wider importance.**

We will revise the conclusions as suggested.

*Line 6: could include examples of the uncertain ice sheet processes you are exploring*

Edited '...more comprehensively explore uncertain ice sheet processes.' To '...more comprehensively sample uncertainties in future climate, ice shelf sensitivity to ocean melting, and their interactions.'

*Line 8-10: I think this sentence would benefit from being rephrased for clarity*

Edited:

"The BISICLES experiments presented here show the important interplay between surface mass balance forcing and ocean driven melt, with high warming, high accumulation forcing conditions leading to mass gain (negative sea level contribution) under low sensitivity to ocean driven melt."

To:

"We present BISICLES experiments showing the important interplay between surface mass balance processes and ocean driven melt in determining Antarctic sea level contribution. Under higher warming scenarios, high accumulation offsets more ocean driven mass loss, when sensitivity to ocean driven melt is low."

*Line 12:"increases sea level contribution by 25 mm" relative to what? I think the whole range of sea level contributions from your experiments should be stated in the abstract.*

Edited to "...increases sea level contribution by 25mm, relative to the no collapse experiments, for both...."

Sentence added on line 12:

" ...accumulation. Overall, we simulate a sea level range across our experiments from X mm to X mm."

**_Line 16: Does this mean "dominated the total sea level change of 20 cm" or ice sheets contributed to 20 cm?_**

Mountain glaciers (41% of total) and thermosteric sea level rise (38% of total) together dominated sea level rise from 1901 to 2018 (Fox-Kemper et al., 2021). Will rephrase sentence beginning line 15 to clarify this:

"...global mean sea level (GMSL), behind thermosteric changes and mountain glaciers (Palerme et al., 2017; Horwath et al., 2022) – which together accounted for 79% of the 20 cm of sea level rise between 1901 and 2018 (Fox-Kemper et al., 2021).

**_Line 36: "future" Antarctic contribution to sea level? Also "represent a greater range of interactions and dynamic processes" reads a bit awkwardly, could it be rephrased, or examples of these "interactions" added?_**

We didn't specify "future" as ice sheet modelling advances have improved characterisation of palaeo sea levels too. However, given the context of the paper, we will add "future" to remove any ambiguity. Suggested edit for improved readability:

Rephrased:
"To better project Antarctic contribution to sea level, ice sheet models have been developed to represent a greater range of interactions and dynamic processes, at higher resolution than ever before, over the past few decades (Pattyn et al., 2017)"

To:
"Ice sheet models are the primary tool for projecting future Antarctic sea level contribution. Over the past few decades, models have developed to represent a greater range of ice sheet processes and climate-ice sheet interactions, at higher resolution than ever before (Pattyn et al., 2017)"

**_Line 44-47: These sentences duplicate the information in the previous paragraph (lines 39-43), I suggest combining and removing the second paragraph of the two._**

As suggested, will merge paragraphs and remove duplicated information i.e.

" ...from the Greenland and Antarctic ice sheets (Nowicki et al., 2016). With a common set of experiments run by different modelling groups, it allows for improved quantification of uncertainty in sea level projections due to choice of ice sheet model."

**_Line 49: I think the estimate of sea level contribution from the ISMIP6 experiments could come earlier/in the first paragraph of the introduction._**

We think it is useful to have some explanation of what ISMIP6 is before giving the results, but we could give the ISMIP6 sea level contribution range in a sentence at the end of line 47.

**Line 60: See later comments and in the general comments section. How does your initial model state here compare to the one included in the initMIP experiments?**

We use the same model initial state as that used in initMIP, we will edit the text to make this clear.

**Line 79: So the model set-up in this paper was exactly the same as the one presented in the initMIP experiments? that is not totally clear from the text. Does that include the choice of basal sliding law, grid resolution, etc? Despite this, I think more space could be given here and in Section 2.3 on how this initial state compares to that of the other models and to observations.**

Yes, we will edit the text to make this clear. As suggested, we will add more detail about the initial state.

**Line 82: A figure of the model mesh would be good in a Supplementary Information document**

The model mesh is adaptive and is simulation dependent i.e. simulations have different meshes, and the mesh evolves through time. However, we are happy to include an example model mesh in the supplementary material.

**Line 85: Perhaps state why this sliding was chosen.**

Add sentence on line 86:
'...coefficient of 0.5. This sliding law accommodates regions of hard beds and slow flow through the Weertman law, and regions of faster flow on deformable beds through the Coulomb law, as well as a smooth transition between the two (equation n., supplementary section n.). Basal traction coefficients...'

**Line 91: How long was this relaxation run? and how does the ice sheet state deviate from the one arrived at after the inversion? Some figures on the change in ice thickness/speed during this relaxation and the comparison to observations at the end of the relaxation/start of the control simulations would be good.**

The relaxation simulation is approximately ten years long. As suggested, we will include figures and text detailing the relaxation simulation, and how it compares to observations in the supplementary material.

**Line 100: The first few sentences of this section duplicate the information already presented in the Introduction. I suggest removing either here or from the introduction (lines 63-66).**

We will remove the duplicate sentences.

**Line 105: Is this the same selection of CMIP6 models included in Payne et al. 2021, if so state this.**

Payne et al. 2021 explore 4 CMIP6 models including CNRM-CM6-1. We chose this model because it had a low emissions scenario available. We will edit the text to make this clearer.

**Line 125: Why did you not use PIGL5th? State this in the text.**

Edited to:
'...are sampled in the simulations presented here (Table 2). With limited time and computational resources, we did not use $PIGL_{5th}$, prioritising instead higher gamma0 simulations to bound the ice sheet sensitivity to ice shelf basal melting.'

**Line 127: Why did you choose to use the surface mass balance from Arthern et al., 2006 rather than RACMO?**

Edited to:
'mass balance from Arthern et al. (2006), following previous BISICLES studies (Cornford et al. 2016) and BISICLES initMIP experiments (Seroussi et al. 2019).'

**Figure 2: The units between the spatial plots and line graph in Figure 1 are the same. It would be good to do that for the SMB anomalies in Figure 2 as well.**

We will edit the figure accordingly.

**Line 141-142: This sentence is almost identical to the one on line 127-128, remove one of them.**

We will edit the text to remove the duplicate sentences.

**Line 142: As mentioned above, it would be good to have more details on your initial model state, the relaxation run, and the subsequent control simulation. It would be useful to include a figure of the surface mass balance and basal melt rates used during the control simulation.**

As suggested, we will include figures and text detailing the relaxation simulation and initial model state. We will also include a figure of the control surface mass balance and basal melt rates for the control simulation in the supplementary information.

**Line 151: None of the place names in this paragraph are labelled on a figure. I guess the assumption is people will know where they are, but it would be nice to label the main ice shelves.**

*Line 158: It's almost impossible to see the grounding line retreat on Fig. 3b, I suggest making the map much larger or even creating an inset(s) for the key regions talked about in this paragraph.*

As suggested, we will edit the figure for clarity and label the main locations mentioned in the text.

*Line 187: Be consistent with use of negative numbers for mass gain/sea level fall. Improve the readability of the following sentence.*

Following reviewer 2 feedback, we will revise all presented sea level contribution results so that the control is not subtracted, but will ensure consistency in revised results.

*Line 194-197: Am I correct in thinking you didn't merge these basins and this was done in Jourdain et al., 2020? If so, just state that you use the basins from Jourdain et al., 2020 and remove the detail here.*

This is correct, we will edit the text accordingly.

*Line 200: It would be nice to see the perturbation experiments alongside the control simulation (without subtracting it). Perhaps a figure in a supplementary information document.*

Following reviewer 2 feedback, we will edit the manuscript to present main results without subtracting the control simulation. For completeness, and to aid comparison with the main ISMIP6 results, we will also present the results with control subtracted in the results table 2 (or supplementary).

*Line 206: Make it clear when you say "Filchner-Ronne has a large area…" you are talking about the catchment not just the ice shelf?*

Edited to "Filchner-Ronne drainage basin has a large area…"

*Line 210: The important compensating effect of accumulation on ocean-driven mass loss has been noted for the Filchner-Ronne region by a number of previous studies, some of which might be worth citing here e.g. Cornford et al. 2015 and Wright et al., 2014.*

We agree that it would be useful to include these references, and will edit the text accordingly.

*Section 3.4: Throughout this section there are almost no references to any figures where the results can be found.*

We will edit the text to add figure references where relevant.

*Figure 5: This figure caption is incomplete. State what each panel shows and add the CMIP forcing to the caption as well.. Also add panel labels to the figure.*

Figure caption edited to:
"Subplots show the grounded area for NorESM1-M (a.), CCSM4 (b.), MIROC-ESM-CHEM (c.) and CNRM-CM6 (d.) experiments from 2015 to 2100"

*Line 224: Could state that an increase in VAF in the Filchner-Ronne and Ross regions in the control simulations is consistent with the present-day trend in VAF and some references e.g. Rignot et al., 2019.*

Edited to: "…VAF throughout the control simulation – broadly consistent with 1979-2019 VAF trend in these regions (Rignot et al., 2019)."

*Line 225: Given that this section of the results is quite short it could just be combined with the basin discussions in the previous section. Figure 9 could probably be removed or moved to a supplement.*

As suggested, we will merge this with the previous section, and move Figure 9 to the supplementary text.

*Line 222: Be careful here, this sentence is quite confusing, you are saying that it is "equivocal" that there is potential for MISI in the ASE, but that there is not evidence that for MISI happening yet. I would rephrase this to improve the clarity.*

Rephrased:
"Evidence for marine ice sheet instability in the ASE is equivocal … incontrovertible evidence of MISI (Fox-Kemper et al., 2021)."

To:
"It is not clear that marine ice sheet instability has been initiated in the ASE, with the IPCC AR6 stating that observed flow regimes in the ASE are compatible with but not incontrovertible evidence of MISI (Fox-Kemper et al., 2021)."

*Figure 7: Needs panel labels and increase the font size in the legend.*

Figure will be edited as suggested.

*Section 4.1. The entirety of this section reads as a stream of new results and reference to new figures. I think this would be better moved to the results. I also don't know how useful the spatial discussion is, and was wondering if Figures 10/11 would be better in a supplement and replaced with a figure that shows integrated ice sheet/regional surface mass balance and dynamic ice discharge instead.*

Following feedback here and from other reviewers, this section will be moved to results. As suggested, we will replace the figures 10 and 11 with a single figure showing regional (EAIS, WAIS and Peninsula) integrated surface mass balance and discharge.

*Figure 8: The y-axis label is "SLC" but throughout the manuscript you use "SLE" I would be consistent with one or the other. Also I can see the reason for a consistent y-axis scale, but it means it some basins it is impossible to see any change through time, so I recommend modifying the y-axis scales individually.*

Axis labels and scales will be changed as suggested.

*Figure 9: I am not sure about showing the SLC lines on top of the spatial plots, they are difficult to see and in several cases they obscure the text. Also the colourbar for the thinning is quite saturated. Could you just show the inland/grounded ice thinning only instead?*

We will edit the plots to only show grounded ice thickness change as suggested, and improve legibility of the line plots and text.

*Line 278: Not sure this should be the start of a new paragraph. Also, is the same true for other unconstrained ice shelves, e.g. Thwaites?*

Yes, sector 9 (ASE) sea level contribution for CNRM relative to control is lower under SSP5.85 than SSP1.26. Will add reference to figure 8.

*Line 279: I did not find a clear statement as to why you did not also use the PIGL sensitivities for these CMIP6 model runs?*

See response to earlier comment on this.

*Figure 10: The caption is incomplete. Refer to experiments in Table 2 (as done in Fig. 9). See earlier suggestions to replace this figure with integrated ice sheet/region wide SMB.*

As suggested, this figure will be replaced with a figure showing regional (EAIS, WAIS and Peninsula) integrated surface mass balance and grounding line discharge.

*Line 296: This section also reads as new results. If it remains as a Discussion section I suggest adding references to more papers that have shown similar results. E.g. for statements about compensating effects of increased precipitation with warming in East Antarctica mass for example Jordan et al., 2023- https://doi.org/10.1038/s41467-023-37553-2 and Stokes et al. 2022 and references therein https://doi.org/10.1038/s41586-022-04946-0).*

As suggested here and by other reviewers, we will move results into the results section. We will include references suggested in edited discussion section.

***Line 305-312: This paragraph again is a list of new results without any reference to any figures/tables where these results can be found. Same is true for the following paragraph.***

As suggested here and by other reviewers, we will move results into the results section, and include figure references where needed.

***Line 356: Given that this equation is not used in this manuscript it feels a bit out of place in the Discussion, I would suggest removing it and just referring to the equation number in that paper instead.***

Lipscomb et al. 2021 do not give the equation separately in their 2021 paper, they state the equation from Jourdain et al. 2020 and write their modification in the text. However, we will remove the equation as suggested, and reference Lipscomb et al. 2021.

***Line 414: There are several references to Sections of Edwards et al., 2021. I think in most cases it would be better to briefly summarise the findings in that paper in your manuscript so the reader doesn't have to frequently go back and forth.***

We will edit the text accordingly.

***Figures 14, 15, 16: The text describing the details of each simulation overlaps large parts of the figure/results, I suggest removing and just directing the reader to Table 2, where the details of the experiments are.***

Following reviewer 1 feedback, we will replace these figures with one showing the experiments mentioned in the main text.

***Line 421: It is my understanding that future work is underway to extend simulations to 2300? is that worth mentioning here?***

Edited sentence 'To confirm this, extending … worthwhile extension on this work'. To 'Work is ongoing to extend these simulations to 2300, and will shed valuable light on mass loss under hight basal melt sensitivity beyond 2100'

***Line 427: Can you state why you omitted the PIGL5 values?***

See earlier comment response.

---

## Author Comment (AC4)

Reviewer 1 ( Dr Zhang) feedback

We thank Dr Zhang for his helpful comments on the first paper draft. Where minor suggested edits have been proposed, these are incorporated into the revised paper and we do not include them below. We respond to more substantial comments in the following. Reviewer comments are bold and italicised for clarity, followed by our response.

***L32-33: I think it is probably better to say something differently between EAIS and WAIS if the SMB change pattern is different, which is in consistent with the experiment design of this paper.***

We will edit these lines to reflect that increased warming increasing snowfall is a more important phenomena for the EAIS.

***L58: put the sentence "BISCILES ISMIP6 ..." in a different paragraph and elaborate the reason you choose BISCILES for this study.***

As suggested, we will include a paragraph break and elaborate our reasons for choosing BISICLES:

BISICLES ISMIP6 experiments were included in the synthesis and sensitivity tests of Edwards et al. (2021). We chose BISICLES to complement the original ISMIP6 ensemble experiments because of it's use of the L1L2 flow approximation, making it well suited to simulating marine ice sheets, and adaptive mesh refinement. This allows BISICLES to capture grounding line dynamics at high resolution, whilst maintaining computational efficiency"

***L85-86: Regarding "m=1/3 and Coulomb friction coefficient", I think you still need put some basic important equations here, like several equations describing the L1L2 approximations, and then you can properly get those model parameters settled somewhere.***

As suggested, we will include a more detailed set of model equations, if not in the main text then in the supplementary section.

***L91: Regarding "the calving front is fixed", but there are also several experiments that you calve all ice shelves away, correct?***

No, collapse experiments only remove shelf area where 10-year average melt exceeds 725 mm/a in CCSM4 – i.e. over limited regions of the shelf. However, we will amend the main text to make this clearer.

***Table 2: Looking at the Collapse On experiments here, it reminds me ABUMIP. Have you compare your results with that of ABUMIP? If not, I suggest doing some***

*analysis. Also, the numbers at the "Sea level contribution" column are not exactly the same as in your following figures (e.g., figure 11), please check.*

Whilst the collapse on experiments remove *some regions* of the ice shelves instantaneously, this is not quite comparable to ABUMIP. ABUMIP removed *all* ice shelves immediately, whereas our collapse on experiments remove regions of shelf ice according to the mask calculated from CCSM4 2m air temperature, calculated using the equation from Trusel et al. 2015 (ref). We will edit the paper to make this clearer to the reader.

*L268-270: I don't think this discussion is necessary here, as SLR is directly contributed by VAF and there is very complex relationship between basal melt of ice shelf and SLR.*

As suggested, we have removed these sentences.

*Figure 10: For SMB, why are all ice shelves are missing? In addition, the spatial pattern here is not clear. Maybe you should try another way to plot them - maybe log scale?*

We have amended the figures to make the shelf edge contours clearer and changed the colour scale for improved clarity.

*L305-312: I think you should say something about the two major ice shelves in WAIS, Filchner Ronne and Ross ice shelf. For example, does the basal melt of Filchner Ronne ice shelf increase nearly proportionally to that of Ross ice shelf?*

We will include discussion of how basal melt co-evolves in time for the two major WAIS ice shelves.

*L325-330: So can we get a conclusion that the buttressing of ice shelf can contribute a 20-30 mm SLR?*

We cannot say this as we do not look at full shelf removal. However, we could make a more caveated statement that 'Based on the temperature-melt relationship proposed in Trusel et al. (2015), and a conservative interpretation of the limit of stability for ice shelves, under CCSM4 temperatures, ice shelf collapse can contribute to 20-30 mm SLR to 2100'.

*Section 4.5: This is probably the major concern of this study. The analysis and figures here seems to be a bit overlapping with previous studies like Seroussi et al. (2020). I doubt if it is necessary to compare BISICLES with all other different types of models. Maybe you can just compare it with other higher order models, which I*

*think makes more sense. Then it might be possible that you can put all comparisons in a single figure, instead of showing them similarly in 3 figures (14-16).*

We have merged the figures into a single figure showing only those experiments mentioned in the main text. Whilst it could make more sense to only compare with other higher order models, other aspects of model set up (e.g. resolution, initialisation, treatment of basal sliding, numerical error) can have a large impact, so we think it may be good to keep the comparison with other models in the figures. As highlighted, some of the comparison does overlap with previous studies. We will therefore edit down the paragraph beginning line 398.

*Figure 13: Please put the thermal forcing curves in a separate plot, which will make a clearer and nicer figure.*

We will edit the figure so that the thermal forcing curves are in a separate plot.

*Section 4.7: It is not clear to me if you have done the hindcast experiment. Please clarify.*

We agree that we have not sufficiently detailed whether we did hindcast (historical) experiments. Other reviewers have also asked for more clarity around the initialisation process and relaxation run. We will clarify the approach of initialising the model with 2007-2010 velocity observations, completing a relaxation run, and the simulating from 2010.

---

## Author Comment (AC5)

Reviewer 2 (Robel) feedback

We are grateful to Dr Robel for reviewing the manuscript and providing thoughtful and constructive feedback. Where minor suggested edits have been proposed, these are incorporated into the revised paper and we do not include them below. We respond to more substantial comments in the following. Reviewer comments are bold and italicised for clarity, followed by our response.

***I do think much of the "discussion" section felt like just further description of results. In particular, I would suggest that section 4.1-4.4 be moved into section 3 since they are mainly a description of the results without much discussion or comparison to other studies. I also think these discussions of the results should probably be condensed by maybe 20-30% for the sake of readability.***

The discussion section reading like more results, particularly section 4.1, is an issue that has also been highlighted by other reviewers. As such, we will edit the discussion so that new results are moved to the appropriate paper section. Moreover, on reflection and following reviewer feedback, we agree that editing the manuscript down would be improve readability. As suggested, we will edit down the discussion of the results in the section highlighted.

***Perhaps my main concern with the science in this paper is how the control run is treated and discussed. I understand that it is standard ISMIP6 practice to subtract the control run from all results such that the resulting numbers represent "sensitivities" of the ice sheet to future emissions forcing. However, this procedure is then somewhat at odds with presenting the results as true projections of future sea level rise, as they are in this paper. While here in this paper and in other ISMIP6 publications the control is often discussed as representing model "drift", it does lump together many potential real sources of ice sheet change including the transient evolution of the initialized ice sheet state, which is out of equilibrium. The paper says as much around lines 220-221 where it says "Whilst subtracting the control can account for model drift, it may also in this instance be removing the sea level signal from ASE's long timescale to retreat initiated before 2015". I think this is quite important because the control run here simulates a non-trivial contribution to sea level rise (6 cm), comparable at first order to the forced changes simulated in the non-control simulations. Thus, when the paper says (e.g.,) that so-and-so simulation represents a "sea level fall", this isn't accurate. Rather, such simulations represents less sea level rise than in the control simulation, but the raw simulation is in fact projecting sea level rise (since even the most "sea level fall" is 5.3 cm, less than the SLR in the control). This can be quite confusing for a reader who is looking to this paper simply for sea level projections. My suggestion would thus be to revise the language throughout the text to discuss the projections as being relative to the sea level rise simulated in the control (i.e., not a "sea level fall" but "less sea level rise than in the control", and not "sea level rise" but "more sea level rise than in the control"). You say something like this for one part of the analysis (line 201), but it applies to all the analyses presented in this paper. Alternatively, you can just not subtract the control run in the plotted results***

*as presented, while still discussing the control run at length. I understand that this is a departure from ISMIP6, but given that we are already moving on to ISMIP7 as a community, this paper could point to a better way to think about considering the control run.*

As noted by the reviewer, we follow the ISMIP6 convention of subtracting a 'control' run from our main projections. However, we are not consistent enough in presenting these results as 'relative to control', which can confuse the reader as pointed out by Dr Robel – e.g. presenting simulations as "sea level fall", when in fact they show smaller sea level rise than in the control. As pointed out in both the reviewer feedback and line 220-221 of the manuscript, subtracting the 'control' does remove some dynamic sea level contribution not primarily driven by model forcing, that is none-the-less an important part of the future sea level contribution in our model. We therefore agree that it may be clearer to not subtract the 'control' simulation in our plotted results. However, for ease of comparison with other ISMIP6 publications, we will include results with control subtracted alongside those with the control not subtracted in table 2 (or supplementary).

**L90: It would make more sense to say that you set the rate factor in effective viscosity purely based on temperature and then you also invert for damage given the A(T) field. (If I understand properly). Not sure if this is different than just inverting for A, but perhaps I don't understand.**

We have made the following edit for clarity:
"Whilst BISICLES uses a depth integrated momentum balance equation, the rate factor A(T) in effective viscosity is based on 3D ice temperature. The inverted parameter phi corrects the vertically integrated effective viscosity in essentially the same way as a damage parameter D (phi = 1 – D), but will conflate the influence of errors in the ice temperature and thickness, as well as the form of the rate factor A(T) (Cornford et al. 2015)".

We have also changed "...the ice damage coefficient are estimated..." to "...the effective viscosity coefficient phi..." on line 86.

**Section 2.2: how is basal melt treated at/across the grounding line?**

The sentence "Basal melting is only applied in cells whose centre is at floatation" will be added to the end of line 125.

**L142: I'm quite confused about why basal melting is applied in this way for the control run? Is it time dependent? Does it vary as the model evolves or is it prescribed at the beginning and then held constant. I'm not sure how I understand the sense in which this is a control. Don't other ISMIP6 models just apply a constant-in-time basal melt forcing for the control?**

The basal melt is time dependent in the sense that it adjusts to remove additional thickening in floating grid cells as this evolved through time (i.e. from advection and SMB). In projection simulations, melt anomalies are applied so that thinning corresponds to the melt anomaly as for BISICLES initMIP experiments. As suggested by other reviewers, we will provide more detail on the control.

**Section 3.1: the paper mainly discussed how the forced simulations compare to other ISMIP6 models, but it would be useful to know how the control simulation compares to them as well**

We will edit the text to mention results of the control simulations compared with other models participating in ISMIP6 (based on table B2 in Seroussi et al. 2020). However, this comparison will be added to section 4.5 'Comparison with other models'.

**L160: is the reason for slow down at major ice shelves the lack of calving in this model?**

**L180: is it possible that the increase in floating area is causing an increase in buttressing. This is an artifact of models that fix the calving front, as discussed in Haseloff and Sergienko 2018, and may have considerable upstream effects on marine ice sheet stability**

We will add discussion of how the fixed front calving may impact buttressing and grounding line dynamics, and slowdown in the control.

**Figure 7: to me it would make more sense to have the AP panel with the same y-axis as the other panels to emphasize the very different scale of contribution, but can understand if the authors would prefer to keep it this way for legibility**

Whilst we can see why this might be helpful for the reasons the reviewer highlights, as noted, at the same scale as the other plots the plot loses legibility. Moreover, positioned as it is on the lower row, we hope the difference in scale compared with the WAIS and EAIS plots will be more obvious to the reader. We will however emphasise the difference in scale on the figure caption.

**L270: I am confused by this sentence since the choice of gamma_0 is independent of the choice of GCM. I can see how the result is dependent both on GCM and gamma_0, but not how one is dependent on the other. Perhaps more explanation is needed.**

We have removed these lines following reviewer one comments.

**L384: It is known that models with friction interpolated across the grounding line/zone are more sensitive and tend to have larger response to forcing than models with more conventional schemes (Tsai et al. 2015). It seems like that is probably playing a role here.**

Edited line 385:
"...in BISICLES (Cornford et al., 2016). Previous studies have also highlighted that models using sub-grid interpolation schemes at the grounding line are more sensitive to forcing than conventional models (Tsai et al., 2015)."

**L390: this is where it would be good to know how basal melt at the grounding line is treated in BISICLES, since this has a big influence on the model sensitivity as Seroussi and Morlighem showed.**

edit line 390:
"core experiments for WAIS, which does not implement a sub-grid interpolation scheme for basal melting (Seroussi and Morlighem, 2018)."

**L395: technically, the sliding scheme in BISICLES is closer to Tsai et al. 2015 than a purely Weertman sliding law, which may be a point of difference with ISSM.**

This is a useful point, edited from 395:

"...than basal sliding law at comparable resolution (Cornford et al. 2020). Whilst BISICLES and ISSM have Weertman sliding over much of the domain, BISICLES uses a Tsai et al. (2015) type sliding law with Coulomb sliding close to the grounding line. This difference could contribute where higher sea level contributions are simulated in BISICLES. The mm-scale magnitude of this difference is comparable to that found in previous studies comparing Weertman-only and Tsai et al. (2015) type sliding laws (Nias et al., 2018, Bowan and Gudmundsson, 2024)"

Bowan paper https://tc.copernicus.org/articles/16/4291/2022/
Nias paper https://agupubs.onlinelibrary.wiley.com/doi/full/10.1002/2017GL076493

**L475: I would hope that the statement "Data is available on request, and will be publicly available in due course" is merely a placeholder for the pre-print, since I'm not sure it is useful for a paper publishing important results contributing to widely used sea level projections. My suggestion would be to make these data available before the paper is published.**

We are finalizing data for upload to Zenodo and will upload before submitting the final manuscript.

---

## Author Response (AR1)

Author response: ISMIP6-based Antarctic Projections to 2100: simulations with the BISICLES ice sheet model

Please find below author response to all reviewer comments, comments are in bold.

**Response to reviewer 1**

***L32-33: I think it is probably better to say something differently between EAIS and WAIS if the SMB change pattern is different, which is in consistent with the experiment design of this paper.***

We have added '…,particularly over East Antarctica,…' to line 35 in the updated manuscript.

***L58: put the sentence "BISCILES ISMIP6 …" in a different paragraph and elaborate the reason you choose BISCILES for this study.***

As suggested, we have included a paragraph break and elaborated on our reasons for choosing BISICLES in sentences on line 61 starting 'We chose BISICILES to complement…'

***L85-86: Regarding "m=1/3 and Coulomb friction coefficient", I think you still need put some basic important equations here, like several equations describing the L1L2 approximations, and then you can properly get those model parameters settled somewhere.***

As suggested, we have added a supplementary information document along with the manuscript that includes the model equations.

***L91: Regarding "the calving front is fixed", but there are also several experiments that you calve all ice shelves away, correct?***

No, collapse experiments only remove shelf area where 10-year average melt exceeds 725 mm/a in CCSM4 – i.e. over limited regions of the shelf. We added a clarifying sentence on line 153 – 'In these experiments, ~30% of the original Bedmap2 ice shelf area is removed by 2100'.

***Table 2: Looking at the Collapse On experiments here, it reminds me ABUMIP. Have you compare your results with that of ABUMIP? If not, I suggest doing some analysis. Also, the numbers at the "Sea level contribution" column are not exactly the same as in your following figures (e.g., figure 11), please check.***

Whilst the collapse on experiments remove *some regions* of the ice shelves instantaneously, this is not quite comparable to ABUMIP. ABUMIP removed *all* ice shelves immediately, whereas our collapse on experiments remove regions of shelf ice

according to the mask calculated from CCSM4 2m air temperature, calculated using the equation from Trusel et al. 2015 (ref). See above edit.

**L268-270: I don't think this discussion is necessary here, as SLR is directly contributed by VAF and there is very complex relationship between basal melt of ice shelf and SLR.**

As suggested, we have removed these sentences.

**Figure 10: For SMB, why are all ice shelves are missing? In addition, the spatial pattern here is not clear. Maybe you should try another way to plot them - maybe log scale?**

Following reviewer 3 comments, we have removed this figure.

**L305-312: I think you should say something about the two major ice shelves in WAIS, Filchner Ronne and Ross ice shelf. For example, does the basal melt of Filchner Ronne ice shelf increase nearly proportionally to that of Ross ice shelf?**

See lines 293-298 in the updated manuscript.

**L325-330: So can we get a conclusion that the buttressing of ice shelf can contribute a 20-30 mm SLR?**

We cannot say this as we do not look at full shelf removal. However, we could make a more caveated statement that 'Based on the temperature-melt relationship proposed in Trusel et al. (2015), and a conservative interpretation of the limit of stability for ice shelves, under CCSM4 temperatures, ice shelf collapse can contribute to 20-30 mm SLR to 2100', which we have added to our conclusions.

**Section 4.5: This is probably the major concern of this study. The analysis and figures here seems to be a bit overlapping with previous studies like Seroussi et al. (2020). I doubt if it is necessary to compare BISICLES with all other different types of models. Maybe you can just compare it with other higher order models, which I think makes more sense. Then it might be possible that you can put all comparisons in a single figure, instead of showing them similarly in 3 figures (14-16).**

We have merged the figures into a single figure showing only those experiments mentioned in the main text. Whilst it could make more sense to only compare with other higher order models, other aspects of model set up (e.g. resolution, initialisation, treatment of basal sliding, numerical error) can have a large impact, so we think it may be good to keep the comparison with other models in the figures. As highlighted, some

of the comparison does overlap with previous studies. We have therefore removed the final paragraph in this section (lines 398-404 in the original manuscript), as it was largely echoing results from elsewhere.

**Figure 13: Please put the thermal forcing curves in a separate plot, which will make a clearer and nicer figure.**

We have edited the figure accordingly.

**Section 4.7: It is not clear to me if you have done the hindcast experiment. Please clarify.**

We have clarified the discussion of our initialisation approach in the Methods section, and added a supplementary document with more details.

**Response to reviewer 2**

*I do think much of the "discussion" section felt like just further description of results. In particular, I would suggest that section 4.1-4.4 be moved into section 3 since they are mainly a description of the results without much discussion or comparison to other studies. I also think these discussions of the results should probably be condensed by maybe 20-30% for the sake of readability.*

As suggested, we moved the discussion sections 4.1-4.3 to the results section, and edited it down by ~20%.

*Perhaps my main concern with the science in this paper is how the control run is treated and discussed. I understand that it is standard ISMIP6 practice to subtract the control run from all results such that the resulting numbers represent "sensitivities" of the ice sheet to future emissions forcing. However, this procedure is then somewhat at odds with presenting the results as true projections of future sea level rise, as they are in this paper. While here in this paper and in other ISMIP6 publications the control is often discussed as representing model "drift", it does lump together many potential real sources of ice sheet change including the transient evolution of the initialized ice sheet state, which is out of equilibrium. The paper says as much around lines 220-221 where it says "Whilst subtracting the control can account for model drift, it may also in this instance be removing the sea level signal from ASE's long timescale to retreat initiated before 2015". I think this is quite important because the control run here simulates a non-trivial contribution to sea level rise (6 cm), comparable at first order to the forced changes simulated in the non-control simulations. Thus, when the paper says (e.g.,) that so-and-so simulation represents a "sea level fall", this isn't accurate. Rather, such simulations represents less sea level rise than in the control simulation, but the raw simulation is in fact projecting sea level rise (since even the most "sea level fall" is 5.3 cm, less than the SLR in the control). This can be quite*

*confusing for a reader who is looking to this paper simply for sea level projections. My suggestion would thus be to revise the language throughout the text to discuss the projections as being relative to the sea level rise simulated in the control (i.e., not a "sea level fall" but "less sea level rise than in the control", and not "sea level rise" but "more sea level rise than in the control"). You say something like this for one part of the analysis (line 201), but it applies to all the analyses presented in this paper. Alternatively, you can just not subtract the control run in the plotted results as presented, while still discussing the control run at length. I understand that this is a departure from ISMIP6, but given that we are already moving on to ISMIP7 as a community, this paper could point to a better way to think about considering the control run.*

As suggested, we have edited the discussion of the control ensemble (see methods section 2.3), and included more detail on the control in the supplementary material. We have also changed our results and figures so that the control is not subtracted, with the exception of the final figure – where this was necessary for comparison with other ISMIP6 submissions, which do subtract the control. For ease of comparison with other ISMIP6 models, we have shown both sets of results in table 2.

*L90: It would make more sense to say that you set the rate factor in effective viscosity purely based on temperature and then you also invert for damage given the A(T) field. (If I understand properly). Not sure if this is different than just inverting for A, but perhaps I don't understand.*

We have made the following edit for clarity:
"Whilst BISICLES uses a depth integrated momentum balance equation, the rate factor A(T) in effective viscosity is based on 3D ice temperature. The inverted parameter phi corrects the vertically integrated effective viscosity in essentially the same way as a damage parameter D (phi = 1 – D), but will conflate the influence of errors in the ice temperature and thickness, as well as the form of the rate factor A(T) (Cornford et al. 2015)".

We have also changed "…the ice damage coefficient are estimated…" to "…the effective viscosity coefficient phi…" on line 86.

*Section 2.2: how is basal melt treated at/across the grounding line?*

The sentence "Basal melting is only applied in cells whose centre is at floatation" has been added to the end of line 137.

*L142: I'm quite confused about why basal melting is applied in this way for the control run? Is it time dependent? Does it vary as the model evolves or is it prescribed at the beginning and then held constant. I'm not sure how I understand the sense in which this is a control. Don't other ISMIP6 models just apply a constant-in-time basal melt forcing for the control?*

We have added more detail on the control in section 2.3, as well as in the supplementary material we have included.

***Section 3.1: the paper mainly discussed how the forced simulations compare to other ISMIP6 models, but it would be useful to know how the control simulation compares to them as well***

We have added comparison with other ISMIP6 model controls in section 4.1 of the updated manuscript.

***L160: is the reason for slow down at major ice shelves the lack of calving in this model?***

***L180: is it possible that the increase in floating area is causing an increase in buttressing. This is an artifact of models that fix the calving front, as discussed in Haseloff and Sergienko 2018, and may have considerable upstream effects on marine ice sheet stability***

Have added some discussion of our fixed front calving here – line 199 in the updated manuscript.

***Figure 7: to me it would make more sense to have the AP panel with the same y-axis as the other panels to emphasize the very different scale of contribution, but can understand if the authors would prefer to keep it this way for legibility***

Whilst we can see why this might be helpful for the reasons the reviewer highlights, as noted, at the same scale as the other plots the plot loses legibility. Moreover, positioned as it is on the lower row, we hope the difference in scale compared with the WAIS and EAIS plots will be more obvious to the reader. We have emphasised the difference in scale on the figure caption.

***L270: I am confused by this sentence since the choice of gamma_0 is independent of the choice of GCM. I can see how the result is dependent both on GCM and gamma_0, but not how one is dependent on the other. Perhaps more explanation is needed.***

We have removed these lines following reviewer one comments.

***L384: It is known that models with friction interpolated across the grounding line/zone are more sensitive and tend to have larger response to forcing than models with more conventional schemes (Tsai et al. 2015). It seems like that is probably playing a role here.***

"...in BISICLES (Cornford et al., 2016). Previous studies have also highlighted that models using sub-grid interpolation schemes at the grounding line are more sensitive to forcing than conventional models (Tsai et al., 2015)." Added to line 386 in new manuscript.

*L390: this is where it would be good to know how basal melt at the grounding line is treated in BISICLES, since this has a big influence on the model sensitivity as Seroussi and Morlighem showed.*

"core experiments for WAIS, which does not implement a sub-grid interpolation scheme for basal melting (Seroussi and Morlighem, 2018)." Added to line 393 in updated manuscript.

*L395: technically, the sliding scheme in BISICLES is closer to Tsai et al. 2015 than a purely Weertman sliding law, which may be a point of difference with ISSM.*

"...than basal sliding law at comparable resolution (Cornford et al. 2020). Whilst BISICLES and ISSM have Weertman sliding over much of the domain, BISICLES uses a Tsai et al. (2015) type sliding law with Coulomb sliding close to the grounding line. This difference could contribute where higher sea level contributions are simulated in BISICLES. The mm-scale magnitude of this difference is comparable to that found in previous studies comparing Weertman-only and Tsai et al. (2015) type sliding laws (Nias et al., 2018, Bowan and Gudmundsson, 2024)" added to line 399.

*L475: I would hope that the statement "Data is available on request, and will be publicly available in due course" is merely a placeholder for the pre-print, since I'm not sure it is useful for a paper publishing important results contributing to widely used sea level projections. My suggestion would be to make these data available before the paper is published.*

Data is available on Zenodo

**Reviewer 3 feedback**

**1. I found the both the Introduction and Discussion sections to be unnecessarily long. There are also some instances of duplicated text. I think the Introduction would benefit from being shortened and more concise in places.**

See response to specific comments

**2. I think the paper would benefit by presenting more detail on the initial ice sheet model state and comparing this to observations and the other model states included in the ISMIP6 experiments. It would be good for the Control simulation section of the Results to include more detail on this, and perhaps an additional figure (see Specific comments).**

See response to specific comments

**3. I have several comments with regards to the Figures. 1) I think the number of figures in the manuscript is a little excessive and I wonder if all of them are absolutely necessary for the main messages of the paper. I think some could be moved to a Supplementary Information document at the least. 2) In many cases the Figure captions are not as detailed as they need to be and the individual panels on almost all figures should be labelled. For both points I've included suggestions in the specific comments below.**

See response to specific comments

**4. The Discussion as mentioned above is very long and in places presents a stream of new results, and often without any reference to Figures/Tables where the results can be found making it difficult to follow. I think it would be good if each section of the Discussion is considered whether it belongs there or in the Results. At the very least I think Section 4.1 should be moved to the Results. In general it would be good to consider what detail is/is not necessary, and I've made some suggestions in the specific comments. Also the Discussion mostly compares the results to those of Edwards et al., 2021 and Seroussi et al., 2020, and could benefit from referencing additional papers to support your findings.**

See response to specific comments, discussion sections have been moved to results as suggested.

**5. The Limitations section could benefit from adding several more. For example how might your results have been different if you had used a different basal melt parameterisation? Also it would be good to mention the choice of sliding law, and how this is a potential source of uncertainty that has not been explored in these experiments. Finally, it would be good to discuss some unaccounted for processes in your experiments, e.g. iceberg calving (fixed ice front), and the SMB-elevation feedback/evolving topography during the simulations (that you do mention on Line 128).**

As suggested, we have extended the limitations section to include those suggested.

**6. I think the Conclusions would benefit from being revised, making sure they are a brief summary of the paper and not introducing any new information, and towards the end providing summary sentences on the key finding(s) of the paper and the wider importance.**

As suggested, we have rewritten the conclusions to conform with the reviewer suggestion.

***Line 6: could include examples of the uncertain ice sheet processes you are exploring***

Edited '...more comprehensively explore uncertain ice sheet processes.' To '...more comprehensively sample uncertainties in future climate, ice shelf sensitivity to ocean melting, and their interactions.' From line 6 in updated manuscript.

**Line 8-10: I think this sentence would benefit from being rephrased for clarity**

Edited:

"The BISICLES experiments presented here show the important interplay between surface mass balance forcing and ocean driven melt, with high warming, high accumulation forcing conditions leading to mass gain (negative sea level contribution) under low sensitivity to ocean driven melt."

To:

"We present BISICLES experiments showing the important interplay between surface mass balance processes and ocean driven melt in determining Antarctic sea level contribution. Under higher warming scenarios, high accumulation offsets more ocean driven mass loss, when sensitivity to ocean driven melt is low." From line 9 in new manuscript.

**Line 12: "increases sea level contribution by 25 mm" relative to what? I think the whole range of sea level contributions from your experiments should be stated in the abstract.**

Edited to "...increases sea level contribution by 25mm, relative to the no collapse experiments, for both...." Line 13 of updated manuscript.

Sentence added on line 13 in updated manuscript:

" ...accumulation. Overall, we simulate a sea level range across our experiments from 2 mm to 178 mm."

**Line 16: Does this mean "dominated the total sea level change of 20 cm" or ice sheets contributed to 20 cm?**

Mountain glaciers (41% of total) and thermosteric sea level rise (38% of total) together dominated sea level rise from 1901 to 2018 (Fox-Kemper et al., 2021). Rephrased sentence beginning line 15 of original manuscript to clarify:

"...global mean sea level (GMSL) change from 1901 to 2018, behind thermosteric changes and mountain glaciers (Palerme et al., 2017; Horwath et al., 2022) – which together accounted for 79% of the 20 cm of sea level rise (Fox-Kemper et al., 2021)." (from line 17 in updated manuscript)

***Line 36: "future" Antarctic contribution to sea level? Also "represent a greater range of interactions and dynamic processes" reads a bit awkwardly, could it be rephrased, or examples of these "interactions" added?***

We didn't specify "future" as ice sheet modelling advances have improved characterisation of palaeo sea levels too. However, given the context of the paper, we will add "future" to remove any ambiguity.

Rephrased:
"To better project Antarctic contribution to sea level, ice sheet models have been developed to represent a greater range of interactions and dynamic processes, at higher resolution than ever before, over the past few decades (Pattyn et al., 2017)"

To:
"Ice sheet models are the primary tool for projecting future Antarctic sea level contribution. Over the past few decades, models have developed to represent a greater range of ice sheet processes and climate-ice sheet interactions, at higher resolution than ever before (Pattyn et al., 2017)" (from line 39 in updated manuscript)

***Line 44-47: These sentences duplicate the information in the previous paragraph (lines 39-43), I suggest combining and removing the second paragraph of the two.***

Removed duplicated information and merged (see paragraph beginning line 39)

***Line 49: I think the estimate of sea level contribution from the ISMIP6 experiments could come earlier/in the first paragraph of the introduction.***

Results of ISMIP6 experiment at start of paragraph beginning line 48 in updated manuscript.

***Line 60: See later comments and in the general comments section. How does your initial model state here compare to the one included in the initMIP experiments?***

Have stated on line 82 of updated manuscript that initial condition same as InitMIP, and added section in supplementary on initialisation approach.

***Line 79: So the model set-up in this paper was exactly the same as the one presented in the initMIP experiments? that is not totally clear from the text. Does that include the choice of basal sliding law, grid resolution, etc? Despite this, I think more space could be given here and in Section 2.3 on how this initial state compares to that of the other models and to observations.***

See response to comment above.

***Line 82: A figure of the model mesh would be good in a Supplementary Information document***

See supplementary figure 1.

**Line 85: Perhaps state why this sliding was chosen.**

'...coefficient of 0.5. This sliding law accommodates regions of hard beds and slow flow through the Weertman law, and regions of faster flow on deformable beds through the Coulomb law, as well as a smooth transition between the two (equation n., supplementary section n.). Basal traction coefficients...' added, line 91 in updated manuscript.

**Line 91: How long was this relaxation run? and how does the ice sheet state deviate from the one arrived at after the inversion? Some figures on the change in ice thickness/speed during this relaxation and the comparison to observations at the end of the relaxation/start of the control simulations would be good.**

Have added details in the supplementary.

**Line 100: The first few sentences of this section duplicate the information already presented in the Introduction. I suggest removing either here or from the introduction (lines 63-66).**

Removed duplicate sentences.

**Line 105: Is this the same selection of CMIP6 models included in Payne et al. 2021, if so state this.**

Payne et al. 2021 explore 4 CMIP6 models including CNRM-CM6-1. We chose this model because it had a low emissions scenario available. See edit on line 116 of updated manuscript.

**Line 125: Why did you not use PIGL5th? State this in the text.**

Edited to:
'...are sampled in the simulations presented here (Table 2). With limited time and computational resources, we did not use $PIGL_{5th}$, prioritising instead higher gamma0 simulations to bound the ice sheet sensitivity to ice shelf basal melting.' Line 136 in updated manuscript.

**Line 127: Why did you choose to use the surface mass balance from Arthern et al., 2006 rather than RACMO?**

Edited to:
'mass balance from Arthern et al. (2006), following previous BISICLES studies (Cornford et al. 2016) and BISICLES initMIP experiments (Seroussi et al. 2019).' Line 141 updated manuscript.

*Figure 2: The units between the spatial plots and line graph in Figure 1 are the same. It would be good to do that for the SMB anomalies in Figure 2 as well.*

Tried this, but Kg m-2 yr-1 seemed more interpretable than Gt m-2 yr-1.

*Line 141-142: This sentence is almost identical to the one on line 127-128, remove one of them.*

Removed duplicate text.

*Line 142: As mentioned above, it would be good to have more details on your initial model state, the relaxation run, and the subsequent control simulation. It would be useful to include a figure of the surface mass balance and basal melt rates used during the control simulation.*

As suggested, have included more detail on initial model state, control simulation and control in supplementary material.

*Line 151: None of the place names in this paragraph are labelled on a figure. I guess the assumption is people will know where they are, but it would be nice to label the main ice shelves.*

*Line 158: It's almost impossible to see the grounding line retreat on Fig. 3b, I suggest making the map much larger or even creating an inset(s) for the key regions talked about in this paragraph.*

Have edited figure 3 following above two comments.

*Line 187: Be consistent with use of negative numbers for mass gain/sea level fall. Improve the readability of the following sentence.*

Following reviewer 2 feedback , have revised all presented sea level contribution results so that the control is not subtracted.

*Line 194-197: Am I correct in thinking you didn't merge these basins and this was done in Jourdain et al., 2020? If so, just state that you use the basins from Jourdain et al., 2020 and remove the detail here.*

Edited text accordingly.

*Line 200: It would be nice to see the perturbation experiments alongside the control simulation (without subtracting it). Perhaps a figure in a supplementary information document.*

Following reviewer 2 feedback, have edited all results and figures to not subtract control, and control alongside.

*Line 206: Make it clear when you say "Filchner-Ronne has a large area..." you are talking about the catchment not just the ice shelf?*

Edited to "Filchner-Ronne drainage basin has a large area..." line 220 updated manuscript.

*Line 210: The important compensating effect of accumulation on ocean-driven mass loss has been noted for the Filchner-Ronne region by a number of previous studies, some of which might be worth citing here e.g. Cornford et al. 2015 and Wright et al., 2014.*

Added references line 227.

*Section 3.4: Throughout this section there are almost no references to any figures where the results can be found.*

Have added figure references where relevant.

*Figure 5: This figure caption is incomplete. State what each panel shows and add the CMIP forcing to the caption as well.. Also add panel labels to the figure.*

Figure caption edited to:
"Subplots show the grounded area for NorESM1-M (a.), CCSM4 (b.), MIROC-ESM-CHEM (c.) and CNRM-CM6 (d.) experiments from 2015 to 2100"

*Line 224: Could state that an increase in VAF in the Filchner-Ronne and Ross regions in the control simulations is consistent with the present-day trend in VAF and some references e.g. Rignot et al., 2019.*

Edited to: "...VAF throughout the control simulation – broadly consistent with 1979-2019 VAF trend in these regions (Rignot et al., 2019)." Line 240 updated manuscript.

*Line 225: Given that this section of the results is quite short it could just be combined with the basin discussions in the previous section. Figure 9 could probably be removed or moved to a supplement.*

As suggested, merged with the previous section, and move Figure 9 to the supplementary text.

*Line 222: Be careful here, this sentence is quite confusing, you are saying that it is "equivocal" that there is potential for MISI in the ASE, but that there is not evidence that for MISI happening yet. I would rephrase this to improve the clarity.*

Rephrased:
"Evidence for marine ice sheet instability in the ASE is equivocal ... incontrovertible evidence of MISI (Fox-Kemper et al., 2021)."

To:
"It is not clear that marine ice sheet instability has been initiated in the ASE, with the IPCC AR6 stating that observed flow regimes in the ASE are compatible with but not incontrovertible evidence of MISI (Fox-Kemper et al., 2021)." Line 238 in updated manuscript.

**Figure 7: Needs panel labels and increase the font size in the legend.**

Figure edited as suggested.

**Section 4.1. The entirety of this section reads as a stream of new results and reference to new figures. I think this would be better moved to the results. I also don't know how useful the spatial discussion is, and was wondering if Figures 10/11 would be better in a supplement and replaced with a figure that shows integrated ice sheet/regional surface mass balance and dynamic ice discharge instead.**

Following feedback here and from other reviewers, this section will be moved to results. As suggested, have replaced figures 10 and 11 with a single figure showing regional (EAIS, WAIS and Peninsula) integrated surface mass balance and discharge.

**Figure 8: The y-axis label is "SLC" but throughout the manuscript you use "SLE" I would be consistent with one or the other. Also I can see the reason for a consistent y-axis scale, but it means it some basins it is impossible to see any change through time, so I recommend modifying the y-axis scales individually.**

Axis labels and scales changed as suggested.

**Figure 9: I am not sure about showing the SLC lines on top of the spatial plots, they are difficult to see and in several cases they obscure the text. Also the colourbar for the thinning is quite saturated. Could you just show the inland/grounded ice thinning only instead?**

Edited plots as suggested, and moved to supplementary.

**Line 278: Not sure this should be the start of a new paragraph. Also, is the same true for other unconstrained ice shelves, e.g. Thwaites?**

Edited out following reviewer two feedback, see edited paragraph from line 264 in updated manuscript.

**Line 279: I did not find a clear statement as to why you did not also use the PIGL sensitivities for these CMIP6 model runs?**

See response to earlier comment on this.

***Figure 10: The caption is incomplete. Refer to experiments in Table 2 (as done in Fig. 9). See earlier suggestions to replace this figure with integrated ice sheet/region wide SMB.***

Figure merged with 11, new figure 9 in updated manuscript.

***Line 296: This section also reads as new results. If it remains as a Discussion section I suggest adding references to more papers that have shown similar results. E.g. for statements about compensating effects of increased precipitation with warming in East Antarctica mass for example Jordan et al., 2023- https://doi.org/10.1038/s41467-023-37553-2 and Stokes et al. 2022 and references therein https://doi.org/10.1038/s41586-022-04946-0).***

As suggested here and by other reviewers, moved to results.

***Line 305-312: This paragraph again is a list of new results without any reference to any figures/tables where these results can be found. Same is true for the following paragraph.***

Moved to results, added figure references where relevant.

***Line 356: Given that this equation is not used in this manuscript it feels a bit out of place in the Discussion, I would suggest removing it and just referring to the equation number in that paper instead.***

Removed equation and added reference to lipscomb (line 345 in updated manuscript).

***Line 414: There are several references to Sections of Edwards et al., 2021. I think in most cases it would be better to briefly summarise the findings in that paper in your manuscript so the reader doesn't have to frequently go back and forth.***

On review decided that format of summarising result then referring to specific section of Edwards paper tells reader result, and allows them to find section in Edwards to confirm/ get more detail.

***Figures 14, 15, 16: The text describing the details of each simulation overlaps large parts of the figure/results, I suggest removing and just directing the reader to Table 2, where the details of the experiments are.***

Following reviewer 1 feedback, replaced these figures with one showing the experiments mentioned in the main text.

***Line 421: It is my understanding that future work is underway to extend simulations to 2300? is that worth mentioning here?***

Edited sentence 'To confirm this, extending ... worthwhile extension on this work'. To 'Work is ongoing to extend these simulations to 2300, and will shed valuable light on

mass loss under hight basal melt sensitivity beyond 2100' line 418 in updated manuscript.

**_Line 427: Can you state why you omitted the PIGL5 values?_**

See earlier comment response.

---

## Referee Report (RR1)

**Review of O'Neill et al. 'ISMIP6-based Antarctic Projections to 2100: simulations with the BISICLES ice sheet model'**

**General comments**

This is my second review of the manuscript by O'Neill et al., which presents new ice sheet model experiments using BISICLES and following the ISMIP6 protocol to make projections of sea level rise from the Antarctic ice sheet by 2100. I grateful to the authors for addressing my previous comments, and those of the other reviewers. The manuscript is now substantially improved. I have a few further specific comments prior to publication, which are outlined below.

**Specific comments**

**Line 9:** This sentence is very similar to the one on line 5 starting "We present simulations", I suggest combining them.

**Line 33:** not clear what would "amplify uncertainty" here, do you mean potential for marine ice sheet instability introduces uncertainty in projections? Perhaps rephrase to improve clarity.

**Line 100:** When the ice is thinner than 1 metre is it fully removed from the model domain (rather than just being set to 1 metre and effectively ignored)? In the high-end simulations that are "collapse-off" how much of the ice shelves end up thinning to 1 m and then being calved off?

**Line 170:** Be consistent between "Ronne-Filchner" here and "Filchner-Ronne" elsewhere in the manuscript.

**Line 176:** Could you say why you see this slowdown at Pine Island? has the grounding line advanced and grounded on the ridge downstream of the current position?

**Figure 3:** suggest adding labels (initials) for Amery ice shelf and Totten Glacier to panels b and c. Also label Pine Island and Thwaites in panel f, as you refer to these several times in the main text.

**Line 202:** Is that really the reason for the slow-down in the control simulation at Pine Island, or is it rather that the melt rates are not high enough close to the grounding line, or the sub-grid parameterisation, or the grounding line had grounded on the ridge and subsequently advanced?

**Figures 5 and 6:** I did not notice this before, but why do all the simulations not start at the same grounded/floating area in 2015?

**Line 219:** "increase their VAF up to -21 mm sea level contribution" reads a bit awkwardly, if saying increases to -21 best to state what this is relative to. What was the VAF contribution in the control?

**Line 222:** Restate the range here.

**Line 223:** Remove "on the one hand" and "on the other".

**Figure 8:** Replace the numbers with letters.

**Line 245:** Suggest changing to "compare simulations using different GCM forcings but with the same ice shelf basal melt sensitivity and emission scenario".

**Figure 9:** Are these cumulative values for SMB integrated over the grounded part of the catchment? Add this detail to the caption.

**Line 303-306:** This sentence is too long. Split into two.

**Line 327:** Add "GCM" before model.

**Line 407:** Is the reason the sea level contribution for basal melt sensitivities is similar because a large proportion of the total ice shelf area has been removed and therefore limited regions where basal melt is still applied?

**Line 415:** The start of the limitations section needs an introductory/topic sentence.

**Line 446:** Could also add something about the models ability to replicate the trend in present-day observations.

**Conclusions:** I am grateful the authors made efforts to improve the conclusions in this version. I still think it could be shorter and more concise. I do not feel you need to go into detailed examples of results for individual climate model forcings.

**Supplement:** supplementary figures should be referred to as "Supplementary Figure 1" or "Fig. S1" so that it is clear to the reader that you are not referring to figures in the main text.

---

## Author Response (AR2)

Response to Review of O'Neill et al. 'ISMIP6-based Antarctic Projections to 2100: simulations with the BISICLES ice sheet model'

We thank Reviewer 3 for their further comments and suggested edits, please find below the point by point response to each of these. Reviewer comments are in bold font and responses are in standard font below.

**Line 9: This sentence is very similar to the one on line 5 starting "We present simulations", I suggest combining them.**

Changed line 9: 'We present BISICLES experiments showing...' to 'Our experiments show...'

**Line 33: not clear what would "amplify uncertainty" here, do you mean potential for marine ice sheet instability introduces uncertainty in projections? Perhaps rephrase to improve clarity.**

Have edited sentence to: 'Under anthropogenic warming, ice loss from marine basins could drive accelerating GMSL contribution in coming decades to centuries (Schlegel et al., 2018; Bulthuis et al., 2019; Lowry et al., 2021; Edwards et al., 2019; DeConto and Pollard, 2016; Golledge et al., 2015; Ritz et al., 2015; Seroussi et al., 2024), with uncertainty around unstable marine ice retreat contributing to uncertainty in future sea level projections (Robel et al., 2019).'

**Line 100: When the ice is thinner than 1 metre is it fully removed from the model domain (rather than just being set to 1 metre and effectively ignored)? In the high-end simulations that are "collapse-off" how much of the ice shelves end up thinning to 1 m and then being calved off?**

This is an error in the submitted manuscript which I have now corrected – floating ice thinner than 10 m thickness is calved. For high end collapse off simulations, maximum loss of floating area to thinning to <10m is ~4%. Have added the below sentence to results section line 198:
'In collapse off experiments, automatic calving of shelf ice thinner than 10 m can result in loss of up to ~4% of initial floating area.'

**Line 170: Be consistent between "Ronne-Filchner" here and "Filchner-Ronne" elsewhere in the manuscript.**

Have edited all occurrences to Filchner-Ronne

**Line 176: Could you say why you see this slowdown at Pine Island? has the grounding line advanced and grounded on the ridge downstream of the current position?**

The Pine Island grounding line remains stable on the ridge, remaining largely static throughout the simulation. Thinning there is fastest at the start of the control

simulation, and slows afterwards. Surface elevation gradient along the main Pine Island trunk gets shallower throughout the simulation period, which is not the case for Thwaites. Have edited from line 176 to:

'The most pronounced ice stream speed up in the control simulation occurs in the Thwaites glacier and its ice shelf (Fig. 3f), in response to grounding line retreat. By contrast, Pine Island glacier maintains its grounding line position and slows down between 2015 and 2100 in the control run (Fig. 3f). Pine Island slow down corresponds with shoaling of ice surface gradients over this period (not shown).'

**Figure 3: suggest adding labels (initials) for Amery ice shelf and Totten Glacier to panels b and c. Also label Pine Island and Thwaites in panel f, as you refer to these several times in the main text.**

Edited Figure and caption accordingly.

**Line 202: Is that really the reason for the slow-down in the control simulation at Pine Island, or is it rather that the melt rates are not high enough close to the grounding line, or the sub-grid parameterisation, or the grounding line had grounded on the ridge and subsequently advanced?**

Based on previous literature, buttressing may be a factor – additional detail added in edit from line 176 (see above) with regards to grounding line.

**Figures 5 and 6: I did not notice this before, but why do all the simulations not start at the same grounded/floating area in 2015?**

Our simulations start in 2010, at which point all areas are the same, however, we plot results for the ISMIP6 interval (2015-2100). We have clarified this in the caption by adding:
'Discrepancies in the initial area here and in Fig. 6 are due to our simulations beginning in 2010, whilst we show 2015 to 2100 results for consistency with ISMIP6.'

**Line 219: "increase their VAF up to -21 mm sea level contribution" reads a bit awkwardly, if saying increases to -21 best to state what this is relative to. What was the VAF contribution in the control?**

Edited to: 'In the Filchner-Ronne sector (Fig. 8-15), fourteen simulations increase their VAF relative to 2015, up to -21 mm sea level contribution, with -11 mm sea level contribution in the control.'

**Line 222: Restate the range here.**

Have added

**Line 223: Remove "on the one hand" and "on the other".**

Removed

**Figure 8: Replace the numbers with letters.**

Whilst it is more usual to use letters for subplots, we feel that since the numbers correspond to sectors, which are used for the basins across ISMIP6 studies, it is simpler for the sake of referencing within the text, and comparing with other publications, to keep numerical subplot labels.

**Line 245: Suggest changing to "compare simulations using different GCM forcings but with the same ice shelf basal melt sensitivity and emission scenario".**

Have edited accordingly.

**Figure 9: Are these cumulative values for SMB integrated over the grounded part of the catchment? Add this detail to the caption.**

Have edited the caption to:
'Total annual grounding line flux (mm SLC, positive values for mass loss) for the West Antarctic ice sheet (WAIS) (a) East Antarctic ice sheet (EAIS) (b) and the Antarctic Peninsula (AP) (e). Annual surface mass balance integrated over the grounded area of each region (mm SLC, so negative values indicate ice sheet mass gain) for WAIS (c), EAIS (d) and the AP (f). Region boundaries are also shown (g). Note that y-scales differ.'

**Line 303-306: This sentence is too long. Split into two.**

Edited to:
'CNRM-CM6-1 has an equilibrium climate sensitivity (ECS) of 4.8∘C (Meehl et al., 2020), similar to MIROC-ESM-CHEM (ECS = 4.7∘C): the highest ECS CMIP5 model sampled in ISMIP6 and discussed in Payne et al. (Payne et al. 2021). However, CNRM-CM6-1 ECS is higher than the remaining CMIP5 models, with CCSM4 and NorESM1-M both having an ECS of 2.9∘C (Flato et al., 2013)'

**Line 327: Add "GCM" before model.**

Changed model to GCM

**Line 407: Is the reason the sea level contribution for basal melt sensitivities is similar because a large proportion of the total ice shelf area has been removed and therefore limited regions where basal melt is still applied?**

Increase in sea level contribution with collapse on is similar for both basal melt sensitivities is the same, but baseline is higher under PIGL95. Have edited for clarity: 'As outlined in Section 3.7, the increase in sea level contribution with collapse on is almost identical for both basal melt sensitivities sampled (MeanAnt50 and PIGL95), though the 'no collapse' baseline is higher under PIGL95.'

**Line 415: The start of the limitations section needs an introductory/topic sentence.**

Added: 'Our work complements the ISMIP6 ensemble, providing high resolution simulations of Antarctica with a physically comprehensive model, and exploring uncertainties beyond the original ISMIP6 protocol. However, limitations remain - these are outlined below.'

**Line 446: Could also add something about the models ability to replicate the trend in present-day observations.**

Added comparison end of first paragraph in section 3.1: 'Average sea level contribution is 0.62 mm yr−1 over this period, compared with an implied rate of 0.41 mm yr−1 from 1992 to 2020 based on observations (Otosaka et al., 2023).'

**Conclusions: I am grateful the authors made efforts to improve the conclusions in this version. I still think it could be shorter and more concise. I do not feel you need to go into detailed examples of results for individual climate model forcings.**

Have edited paragraph beginning line 462 to remove example.

**Supplement: supplementary figures should be referred to as "Supplementary Figure 1" or "Fig. S1" so that it is clear to the reader that you are not referring to figures in the main text.**

Figure names and references in text edited accordingly.

---

## Author Response (AR3)

Author's response, O'Neill et al. 'ISMIP6-based Antarctic Projections to 2100: simulations with the BISICLES ice sheet model'

Following discussion with the editor and The Cryosphere figure guidelines, we have followed Reviewer 3 feedback on figure 8 and added letter labels to each subplot, and edited references to this in the main text. In addition, we have changed figure subplot labels from e.g. a. to (a) to follow The Cryosphere figure guidelines.

We thank Reviewer 3 and the Editor for their helpful feedback on this final version of the manuscript.